# SPARSEGEOHOPCA: A GEOMETRIC SOLUTION TO SPARSE HIGHER-ORDER PCA WITHOUT COVARIANCE ESTIMATION

## ABSTRACT

This paper proposes sparseGeoHOPCA, a geometric framework for sparse higher-order principal component analysis (SHOPCA). The method unfolds the input tensor along each mode and reformulates the resulting subproblems as binary linear programs, transforming the nonconvex sparse objective into a tractable geometric form. This eliminates covariance estimation and iterative deflation, leading to improved efficiency and interpretability in high-dimensional and unbalanced settings. Theoretical equivalence with the original SHOPCA formulation is established, and error bounds linked to PCA residuals are derived, providing data-dependent guarantees. The algorithm has total complexity $O(\sum_{n=1}^{N}(k_n^3 + J_n k_n^2))$ per iteration, scaling linearly with tensor size. Extensive experiments demonstrate accurate sparse support recovery, stable classification under 10× compression, high-quality ImageNet reconstruction, and semantic reduction, highlighting robustness and versatility.

## 1 INTRODUCTION

In this paper, we study the sparse higher-order principal component analysis (SHOPCA) problem. Higher-order principal component analysis (HOPCA), or multilinear principal component analysis (MPCA), refers to the extension of classical principal component analysis (PCA) to tensor-structured data, enabling dimensionality reduction and pattern extraction from higher-order data structures (Kolda & Bader, 2009; Lu et al., 2008).

To address the limitations of traditional HOPCA in high-dimensional settings, the introduction of sparsity constraints has emerged as an effective strategy. The motivation for incorporating sparsity is fourfold: (1) Sparsity enhances interpretability by ensuring that each principal component involves only a small subset of relevant features, making the results more understandable and visually interpretable. (2) Sparse representations promote automatic feature selection and improve compression efficiency by focusing on the most informative variables. (3) In high-dimensional, low-sample-size scenarios, sparsity mitigates the statistical instability of classical PCA solutions, improving the accuracy of the estimation (Johnstone & Lu, 2009). (4) SHOPCA preserves the structural integrity of the original tensor while focusing on the most informative subset of features, making it suitable for complex real-world applications such as multimodal learning (Sun et al., 2022), biomedical analysis (Allen, 2012), and recommender systems (Frolov & Oseledets, 2017).

However, introducing sparsity into tensor PCA results in a non-convex combinatorial optimization problem, which is generally NP-hard (Choo & d'Orsi, 2021; Hillar & Lim, 2013). As a result, various approximate algorithms and optimization strategies have been proposed. The seminal contributions by Lu et al. (2006; 2008) pioneered MPCA by decomposing tensor data via mode-wise matrix PCA, laying the foundation for multilinear analysis of high-dimensional data. Building on this, Allen (2012) introduced two influential models, sparse HOSVD and sparse CP, that were among the first to incorporate sparsity-promoting penalties into the tensor PCA framework. These models enabled the discovery of interpretable low-rank structures while simultaneously performing feature selection. Subsequently, (Lai et al., 2014) proposed the multilinear sparse PCA (MSPCA) method, which extended sparse PCA ideas to tensor-valued data and demonstrated effectiveness in applications such as face recognition. While these methods have shown notable success in image and video

analysis (Liu et al., 2018), brain signal processing (Xu et al., 2024a;b), biomedical data interpretation (Zhou et al., 2016), and large-scale sensor networks (Rajesh & Chaturvedi, 2021; Zhang et al., 2021), they often fall short in handling computational efficiency across different tensor modes. In particular, the construction and manipulation of large covariance matrices in high-dimensional settings result in significant memory and computational burdens.

In this paper, we propose a geometry-aware framework for efficiently approximating SHOPCA. Our framework facilitates the use of existing optimization solvers in a plug-and-play manner, allowing flexible and modular integration. Specifically, the framework comprises three main stages: (i) *Tensor Preparation*: Given a SHOPCA problem, we unfold the tensor along each mode and formulate the corresponding sparse matrix PCA subproblems. Initial support sets are selected for each mode-unfolded matrix. (ii) *Geometric Solver for Subproblems*: For each mode-unfolded matrix, we reformulate the subproblem from a geometric perspective and solve a structured binary linear program to identify the most significant columns under sparsity constraints. (iii) *Solution Construction*: We construct the core tensor and factor matrices by combining the solutions obtained across modes, yielding the final decomposition result. These steps are illustrated in Figure 1.

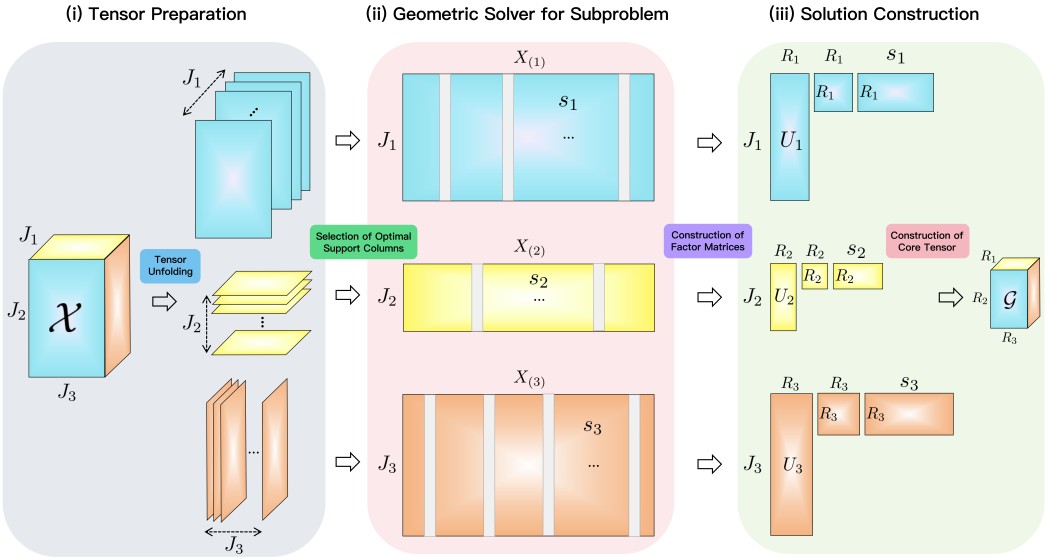

Figure 1: Illustration of the proposed *sparseGeoHOPCA* pipeline on a third-order tensor. (i) The tensor is unfolded along each mode to obtain matricized views. (ii) A geometry-aware solver selects sparse column supports from each mode-wise unfolding. (iii) Based on the selected supports, factor matrices and the core tensor are constructed to yield a sparse multilinear decomposition.

Compared to prior approaches, our framework avoids explicit computation of large-scale covariance matrices, which are often infeasible in high-dimensional settings where the number of columns greatly exceeds the number of rows. This substantially improves computational efficiency while maintaining strong approximation guarantees. The proposed method is more likely to find globally optimal solutions and achieve effective control of approximation error.

We theoretically justify the reformulation of the geometric subproblem in Section 3.2. The worst-case error bounds are derived in Section 3.3, and computational complexity is analyzed in Section 3.4. Extensive empirical evaluations in Section 4 validate the effectiveness of the proposed method.

**Our contributions.** We summarize our contributions as follows:

- We propose *sparseGeoHOPCA*, a novel framework that introduces a geometric perspective to the SHOPCA problem. By formulating mode-wise column selection subproblems from a geometric viewpoint, we transform the original nonconvex sparse optimization into binary linear optimization problems. This approach avoids explicit estimation of covariance matrices and iterative defla-

tion procedures, offering notable advantages in both computational efficiency and interpretability, especially in high-dimensional, unbalanced tensor settings.

- We theoretically establish the equivalence between the proposed geometric subproblems and the original sparse objective, and derive worst-case error bounds for SHOPCA based on residuals from classical PCA. These results provide interpretable, data-dependent guarantees on the approximation quality of sparse projections.

- Our method significantly reduces both computational and memory overhead in high-dimensional regimes. By adopting an alternate optimization strategy and solving the selection of structured support through geometric pruning for each tensor mode, the algorithm achieves a total computational complexity of $O(\sum_{n=1}^{N}(k_n^3 + J_n k_n^2))$ per iteration. This complexity scales linearly with the tensor size, ensuring strong scalability for large-scale tensor applications.

- We perform extensive evaluations on support recovery, classification, and image reconstruction tasks. Experimental results demonstrate that *sparseGeoHOPCA* achieves accurate support selection, maintains classification performance under 10× compression, yields high-quality image reconstruction on ImageNet, and performs effective semantic reduction, highlighting its effectiveness, structural robustness, and generalization beyond tensor-specific applications.

## 2 PROBLEM SETTING AND MOTIVATION

In this section, we provide a formal definition of the sparse higher-order principal component analysis (SHOPCA) problem considered in this work and present the core motivation for integrating geometric feature selection techniques into a tensor-based decomposition framework.

### 2.1 NOTATIONS AND DEFINITIONS

Unless stated otherwise, we adopt the following notation: scalars are denoted by lowercase letters (e.g., $a, b$), vectors by bold lowercase letters (e.g., $\mathbf{v}$), matrices by uppercase letters (e.g., $M$), and tensors by calligraphic letters (e.g., $\mathcal{T}$).

Let $\mathcal{X} \in \mathbb{R}^{J_1 \times J_2 \times \cdots \times J_N}$ be an $N$-th order tensor. Its mode-$n$ matricization, denoted by $X_{(n)} \in \mathbb{R}^{J_n \times \prod_{i \neq n} J_i}$, rearranges the mode-$n$ fibers of $\mathcal{X}$ into columns via the unfolding operator $\mathrm{unfold}_n(\mathcal{X})$. The inverse operation $\mathrm{fold}_n(\cdot)$ reconstructs the tensor from its matricized form, satisfying $\mathcal{X} = \mathrm{fold}_n(X_{(n)})$. The mode-$k$ product (also known as the Tucker product) of $\mathcal{X}$ with a matrix $U_k \in \mathbb{R}^{J_k \times R_k}$ is denoted by $\mathcal{Y} = \mathcal{X} \times_k U_k$, and produces a tensor of size $\mathbb{R}^{J_1 \times \cdots \times J_{k-1} \times R_k \times J_{k+1} \times \cdots \times J_N}$. This transformation projects the mode-$k$ fibers of $\mathcal{X}$ onto a lower-dimensional subspace, and its matrix representation is given by: $Y_{(k)} = U_k X_{(k)}$, where $Y_{(k)}$ is the mode-$k$ matricization of the resulting tensor $\mathcal{Y}$.

### 2.2 HIGHER-ORDER PRINCIPAL COMPONENTS ANALYSIS

Higher-order principal components analysis (HOPCA) extends the classical PCA to tensors via Tucker decomposition. Given $\mathcal{X} \in \mathbb{R}^{J_1 \times \cdots \times J_N}$, HOPCA approximates the tensor using:

$$\mathcal{X} \approx \mathcal{G} \times_1 U_1 \times_2 U_2 \cdots \times_N U_N, \tag{1}$$

where $\mathcal{G} \in \mathbb{R}^{R_1 \times \cdots \times R_N}$ is the core tensor and $U_n \in \mathbb{R}^{J_n \times R_n}$ are orthonormal factor matrices. In component-wise form:

$$\mathcal{X}(i_1, \ldots, i_N) = \sum_{\alpha_1=1}^{R_1} \cdots \sum_{\alpha_N=1}^{R_N} \mathcal{G}(\alpha_1, \ldots, \alpha_N) \prod_{n=1}^{N} U_n(i_n, \alpha_n). \tag{2}$$

This representation drastically reduces the storage and computation cost from $\mathcal{O}(J_1 \cdots J_N)$ to $\mathcal{O}(R_1 \cdots R_N + \sum_{n=1}^{N} J_n R_n)$, offering significant advantages when $J_n \gg R_n$. Optimization is typically achieved using higher-order SVD or alternating least squares (ALS), enabling an effective and interpretable multilinear dimensionality reduction.

## 2.3 SPARSE HIGHER-ORDER PRINCIPAL COMPONENTS ANALYSIS

To enhance interpretability and robustness, sparse PCA has been extended to tensors via the SHOPCA framework. Let $\mathcal{X} \in \mathbb{R}^{J_1 \times \cdots \times J_N}$ be the data tensor and $U_n \in \mathbb{R}^{J_n \times R_n}$ the projection matrix for each mode. The objective is to minimize the projection error while enforcing sparsity and orthogonality as follows:

$$f(U_1, \ldots, U_N) = \|\mathcal{X} - \mathcal{X} \times_1 U_1 U_1^\top \cdots \times_N U_N U_N^\top\|_F^2. \tag{3}$$

The SHOPCA problem is thus formalized as follows:

$$\begin{aligned} \underset{U_1,\ldots,U_N}{\text{minimize}} \quad & f(U_1, \ldots, U_N) \\ \text{subject to} \quad & \|U_n\|_0 \leq k_n, \quad U_n^\top U_n = I_{R_n}, \quad \text{for } n = 1, \ldots, N, \end{aligned} \tag{4}$$

where the $\ell_0$ constraint enforces shared row-sparsity across columns of $U_n$.

**Motivating Idea of This Paper.** The SHOPCA problem is inherently challenging due to its non-convex nature and lack of a closed-form solution. Inspired by the alternating optimization strategy in HOPCA, we decompose the problem into $N$ independent subproblems (see Theorem 3.1). Fixing all $U_m$ for $m \neq n$, the objective simplifies to the following sparse matrix approximation:

$$\begin{aligned} \underset{U_n}{\text{minimize}} \quad & \|X_{(n)} - U_n U_n^\top X_{(n)}\|_F^2 \\ \text{subject to} \quad & \|U_n\|_0 \leq k_n, \quad U_n^\top U_n = I_{R_n}, \end{aligned} \tag{5}$$

where $X_{(n)}$ is the mode-$n$ unfolding of $\mathcal{X}$. In practice, when $J_n \ll \prod_{i \neq n} J_i$, traditional sparse PCA methods relying on covariance estimation become inefficient.

To overcome this bottleneck, we integrate the geometry-aware method into our tensor framework. It circumvents the need for covariance computation and iterative deflation by directly selecting the columns of $X_{(n)}$ with the largest Frobenius norms. The column selection problem is formulated as follows:

$$\begin{aligned} \underset{\mathbf{s} \in \{0,1\}^{\prod_{i \neq n} J_i}}{\text{maximize}} \quad & \sum_{j=1}^{\prod_{i \neq n} J_i} s_j \|X_{(n)}(:,j)\|_F^2 \\ \text{subject to} \quad & \mathbf{e}^\top \mathbf{s} \leq k_n, \quad \forall \sigma^n \subset [\prod_{i \neq n} J_i], \ \eta(\mathbf{s}^{\sigma^n}) > \eta \Rightarrow \sum_{j \in \sigma^n} s_j \leq |\sigma^n| - 1, \end{aligned} \tag{6}$$

where $\eta(\mathbf{s}) = \|X_{(n)}(:,\mathbf{s}) - U_n[\mathbf{s}] U_n[\mathbf{s}]^\top X_{(n)}(:,\mathbf{s})\|_F^2$ measures the reconstruction error in the selected column subset and $U_n[\mathbf{s}]$ denotes the optimal projection basis obtained by solving $U_n[\mathbf{s}] = \arg\max_{U \in \mathbb{R}^{J_n \times R_n}} \text{tr}(U^\top X_{(n)}(:,\mathbf{s}) X_{(n)}(:,\mathbf{s})^\top U)$ subject to $U^\top U = I_{R_n}$. Here, $\mathbf{e} \in \mathbb{R}^{\prod_{i \neq n} J_i}$ is an all-one vector that enforces the sparsity constraint, and $[\prod_{i \neq n} J_i]$ denotes the index set $\{1, 2, \ldots, \prod_{i \neq n} J_i\}$ of all columns in $X_{(n)}$. Each subset $\sigma^n$ represents a candidate column group that is excluded if its reconstruction error exceeds the threshold $\eta$. Details are discussed further in Theorem 3.2.

Although the geometry-aware method was originally introduced for matrix-based sparse PCA, our main contribution lies in its adaptation to the multilinear tensor setting. This integration yields a scalable, non-iterative, and interpretable approach for sparse tensor decomposition, particularly effective in high-dimensional and unbalanced scenarios.

# 3 OUR APPROACH: SPARSEGEOHOPCA

In this section, we propose *sparseGeoHOPCA*, a geometry-aware framework to solve the SHOPCA problem. The method combines alternating optimization with structured column selection inspired by GeoSPCA, enabling efficient and interpretable sparse tensor decomposition.

Section 3.1 outlines the overall algorithm. Section 3.2 details the theoretical foundation, including mode-wise decoupling and sparse PCA reformulation. Section 3.3 provides approximation error bounds. Section 3.4 discusses the computational complexity.

## 3.1 SPARSEGEOHOPCA

In this section, we present the proposed algorithmic framework, sparseGeoHOPCA (Geometry-Aware Sparse Higher-Order PCA), designed to solve the sparse higher-order principal component analysis (SHOPCA) problem defined in 4.

Given an input tensor $\mathcal{X} \in \mathbb{R}^{J_1 \times J_2 \times \cdots \times J_N}$, target Tucker ranks $(R_1, \ldots, R_N)$, sparsity parameters $\{k_n\}_{n=1}^N$, a tolerance threshold $\eta$, and optional initial exclusion sets for each mode, our goal is to identify sparse orthonormal factor matrices $\{U_n\}_{n=1}^N$ and a compact core tensor $\mathcal{G}$.

To this end, we employ an alternating optimization strategy that decouples the problem into $N$ independent subproblems, each corresponding to one tensor mode. For the $n$-th mode, we unfold the tensor into its mode-$n$ matricization $X_{(n)}$ and formulate a binary linear optimization (BLO) problem $\phi^{(n)}$ to select a support vector $\mathbf{s}^0$ that determines a subset of dominant columns.

At each iteration $t$, we compute the approximation error $\|X_{(n)}(:, \mathbf{s}^t) - U_n[\mathbf{s}^t]U_n[\mathbf{s}^t]^\top X_{(n)}(:, \mathbf{s}^t)\|_F^2$. If this error exceeds the tolerance $\eta$, we refine the BLO problem by adding a constraint to exclude the current support $\sigma^t$, and resolve $\phi^{(n)}$ to obtain a new support $\mathbf{s}^{t+1}$. This process continues until the approximation error falls below the threshold. Moreover, whenever a candidate support $s_0$ achieves a smaller error $\eta(s_0) < \eta$, the tolerance is adaptively reset to $\eta(s_0)$ and enforced in subsequent iterations. Once all sparse factor matrices $\{U_n\}_{n=1}^N$ are obtained, the Tucker core tensor is computed as: $\mathcal{G} = \mathcal{X} \times_1 U_1^\top \times_2 U_2^\top \cdots \times_N U_N^\top$.

The complete procedure of the sparseGeoHOPCA algorithm is summarized in Algorithm 1.

---

**Algorithm 1:** sparseGeoHOPCA

**Input:** Tensor $\mathcal{X} \in \mathbb{R}^{J_1 \times J_2 \times \cdots \times J_N}$; target ranks $(R_1, \ldots, R_N)$; sparsity levels $\{k_n\}_{n=1}^N$; tolerance parameter $\eta$; initial support sets $s^*$.

**Output:** Sparse factor matrices $\{U_n\}_{n=1}^N$ and core tensor $\mathcal{G}$.

1 **for** $n = 1$ *to* $N$ **do**

2      Matricize $\mathcal{X}$ along mode $n$ to obtain $X_{(n)} \in \mathbb{R}^{J_n \times \prod_{i \neq n} J_i}$.

3      Formulate the BLO problem $\phi^{(n)}$ below and solve to obtain the support vector $s^0$,

$$\max_{\mathbf{s} \in \{0,1\}^{\prod_{i \neq n} J_i}} \sum_{i=1}^{\prod_{i \neq n} J_i} s_i \|X_{(n)}(:, i)\|_F^2, \quad \text{s.t.} \quad \mathbf{e}^\top \mathbf{s} \leq k_n.$$

     Compute $U_n[s^0]$ by applying PCA to $X_{(n)}(:, s^0)$.

4      **while** $\|X_{(n)}(:, \mathbf{s}^t) - U_n[\mathbf{s}^t]U_n[\mathbf{s}^t]^\top X_{(n)}(:, \mathbf{s}^t)\|_F^2 > \eta$ **do**

5          Add constraint $\sum_{i \in \sigma_t^n} s_i \leq |\sigma_t^n| - 1$ to $\phi^{(n)}$.

6          Re-solve $\phi^{(n)}$ to obtain new support $\mathbf{s}^{t+1}$.

7          Compute $U_n[\mathbf{s}^{t+1}]$ via PCA on $X_{(n)}(:, \mathbf{s}^{t+1})$.

8      Set $U_n \in \mathbb{R}^{J_n \times R_n}$ as the final solution for mode $n$.

9 Compute the core tensor as: $\mathcal{G} = \mathcal{X} \times_1 U_1^\top \times_2 U_2^\top \cdots \times_N U_N^\top$.

---

**Remark 3.1** *In practice, we initialize the support sets $s^*$ using the top-norm (largest $\ell_2$ columns) strategy and set the tolerance $\eta \in [10^{-4}, 10^{-2}]$. A detailed sensitivity analysis of different initialization strategies and parameter choices is provided in Appendix J.*

## 3.2 FRAMEWORK OF *sparseGeoHOPCA*

To address the problem of sparse higher-order principal component analysis (SHOPCA) defined in 4, we propose a geometry-aware alternating optimization framework named *sparseGeoHOPCA*. This method iteratively decouples the full tensor optimization problem into a sequence of sparse subproblems, each corresponding to a single tensor mode. The decomposition is formally established in the following result.

**Theorem 3.1** *Let $(U_1, \ldots, U_{n-1}, U_{n+1}, \ldots, U_N)$ be fixed. Then optimization of $U_n$ in 4 reduces to the sparse matrix approximation problem given in 5.*

*Proof.* See Appendix A. □

To solve this subproblem, we adopt a column selection strategy inspired by the geometry-aware method (Bertsimas & Kitane, 2022). The key idea is to transform the original sparse optimization into a binary linear program with geometric exclusion constraints, which can be efficiently solved using standard integer programming techniques. The following theorem justifies this reformulation.

**Theorem 3.2** *Let $s^0$ be an optimal solution to problem 5. Then there exists a constant $\delta > 0$ such that for any $\eta \in [\eta(\mathbf{s}^0), \eta(\mathbf{s}^0) + \delta]$, any optimal solution to the selection problem in 6 is also an optimal solution to problem 5.*

*Proof.* See Appendix B. □

**Remark 3.2** *The idea of simultaneously optimizing the joint sparse structures across multiple modes is closely related to our current mode-wise formulation. In fact, similar joint formulations have been explored in the literature (Bertsimas & Kitane, 2022), and our approach can be extended along the same lines. For clarity and focus, we do not elaborate on this extension here, but note that it is conceptually aligned with our framework.*

### 3.3 WORST-CASE UPPER BOUND

Although the alternating optimization framework of *sparseGeoHOPCA* guarantees convergence to a locally optimal solution, it remains crucial to understand the quality of the solution obtained in each mode. In particular, we are interested in quantifying how well the sparse projection subspace captures the original data variance compared to its dense (classical PCA) counterpart. To this end, we establish a worst-case upper bound on the reconstruction error $\eta(\mathbf{s}^0)$ obtained from solving the sparse subproblem 6. This bound can be computed a priori using the residual matrix from classical PCA, which serves as an easily accessible upper reference.

**Theorem 3.3** *Let $(\mathbf{s}^0, U_n[\mathbf{s}^0])$ be an optimal solution to Problem 6. Consider the classical PCA solution $V^* \in \underset{V \in \mathbb{R}^{J_n \times R_n}}{\arg\min} \|X_{(n)} - VV^\top X_{(n)}\|_F$ subject to $V^\top V = I_{R_n}$, and define the associated residual matrix as $\epsilon^n = X_{(n)} - V^* V^{*\top} X_{(n)}$. Let $\sigma^n \subset [\prod_{i \neq n} J_i]$ denote the indices corresponding to the $k_n$ columns of $\epsilon^n$ with the highest norm. Then, the following inequality holds:*

$$\eta(\mathbf{s}^0) \leq \|S^0 \epsilon^n\|_F^2 \leq \sum_{i \in \sigma^n} \|\epsilon_i^n\|_F^2, \tag{7}$$

*where $\epsilon_i^n$ is the $i$-th column of the residual matrix $\epsilon^n$, and $S^0 = \mathrm{diag}(\mathbf{s}^0)$. Moreover, the solution $\mathbf{s}^0$ is feasible for problem 6 when $\eta = \|S^0 \epsilon^n\|_F^2$.*

*Proof.* See Appendix C. □

The residual matrix $\epsilon^n$ can be efficiently computed using classical PCA, making the bound on $\eta(\mathbf{s}^0)$ both practical and interpretable. The result provides a data-dependent certificate of approximation quality under sparsity constraints. We now extend this worst-case characterization to the full multilinear setting. The following result provides a global upper bound on the total approximation error of the *sparseGeoHOPCA* decomposition in terms of the mode-wise PCA residuals.

**Theorem 3.4** *Let $\mathcal{X} \in \mathbb{R}^{J_1 \times \cdots \times J_N}$ be an $N$-mode tensor, and let $\{U_n\}_{n=1}^N$ be the sparse projection matrices computed by Algorithm 1. Then, the total reconstruction error $f(U_1, U_2, \ldots, U_N)$ in 4 is upper bounded by the sum of leading residual energies from classical PCA applied to the mode-$n$ unfoldings $X_{(n)}$ of $\mathcal{X}$:*

$$f(U_1, U_2, \ldots, U_N) \leq \sum_{n=1}^N \sum_{i \in \sigma^n} \|\epsilon_i^n\|^2, \tag{8}$$

*where $\epsilon^n = X_{(n)} - V^n V^{n\top} X_{(n)}$ is the residual matrix from classical PCA, and $\sigma^n \subset [\prod_{i \neq n} J_i]$ contains the indices of the $k_n$ columns of $\epsilon^n$ with the highest norm.*

*Proof.* See Appendix D. □

### 3.4 COMPLEXITY ANALYSIS

The computational complexity of Algorithm 1 is primarily determined by its alternating optimization framework. In particular, the computational bottleneck lies in solving the subproblem formulated in problem 6. The complexity of this subproblem can be analyzed in two parts: (i) the total number of generated cutting planes (cuts), and (ii) the computational cost per iteration. In the worst case, the number of cuts can grow exponentially.

Each iteration involves two main operations: (1) solving a binary linear optimization (BLO) problem, and (2) executing a separation oracle to detect violated constraints. The separation step requires solving a classical PCA problem on a matrix of dimension $J_n \times k_n$, which can be efficiently handled by using singular value decomposition (SVD) with a computational cost of $O(k_n^3 + J_n k_n^2)$. According to the analysis in Bertsimas & Kitane (2022), the BLO problem can be solved using a tree search algorithm with a worst-case complexity of $O(k_n)$(see Appendix E for more details.). Therefore, the per-iteration complexity of solving Problem 6 is dominated by the cost of the SVD and is approximately $O(k_n^3 + J_n k_n^2)$. Although the total number of iterations could be exponential in theory, empirical evidence shows that the algorithm typically converges to high-quality solutions within a manageable number of iterations, even for moderately large problem sizes. Summing across all tensor modes, the overall computational complexity of Algorithm 1 is $O(\sum_{n=1}^{N}(k_n^3 + J_n k_n^2))$ per iteration.

Furthermore, it is important to highlight that our method avoids explicit computation of covariance matrices. This design choice not only reduces computational overhead but also significantly reduces memory consumption, particularly in high-dimensional scenarios where $J_n \ll \prod_{i \neq n} J_i$. As a result, problem 6 becomes especially attractive for large-scale tensor applications.

## 4 EMPIRICAL RESULTS

This section presents both synthetic and real-data experiments to evaluate the effectiveness of the proposed *sparseGeoHOPCA*(Geo for short in this section) framework. We begin with support recovery in controlled simulations and then evaluate classification performance under high-dimensional compression. We further validate the method through an ImageNet image reconstruction task, where Geo outperforms matrix-based baselines in both visual quality and computational efficiency. All experiments were conducted on a workstation equipped with an Intel(R) Core(TM) i7-10700 CPU @ 2.90GHz, an NVIDIA RTX 4070 Super GPU, and 64GB RAM. The proposed Geo algorithm was implemented in Python using PyTorch 2.6 and Gurobi 10.0.1 for solving the BLO subproblems. The initialization of support sets and tolerance parameters follows the settings described in Remark 3.1.

### 4.1 SYNTHETIC EXPERIMENTS

We evaluate the support recovery performance of Geo through controlled synthetic simulations based on a low-rank third-order tensor model, under varying sparsity and dimensionality conditions. The observed tensor is generated as $\mathcal{X} = \sum_{k=1}^{K} d_k \, \mathbf{u}_k \circ \mathbf{v}_k \circ \mathbf{w}_k + \mathcal{E}, \quad \mathcal{E}_{i,j,l} \overset{\text{iid}}{\sim} \mathcal{N}(0,1)$, where $K = 1$ and $d_1 = 100$. We consider four simulation scenarios with varying tensor dimensions and sparsity structures (details provided in Appendix F.1). The results demonstrate that our method is particularly effective in accurately identifying sparse structures, significantly outperforming baseline HOPCA in both precision and robustness.

Figure 2 shows the ROC curves averaged over 50 replicates in Scenarios 1 and 2, where only mode-$u_1$ is sparse. In both settings, Geo consistently achieves higher true positive rates and lower false positive rates compared to baseline methods, demonstrating robust support recovery under moderate sparsity and dimensional imbalance.

Appendix F.2 includes ROC curves for Scenarios 3 and 4, and Appendix F.3 reports true/false positive rate comparisons between HOPCA and Geo.

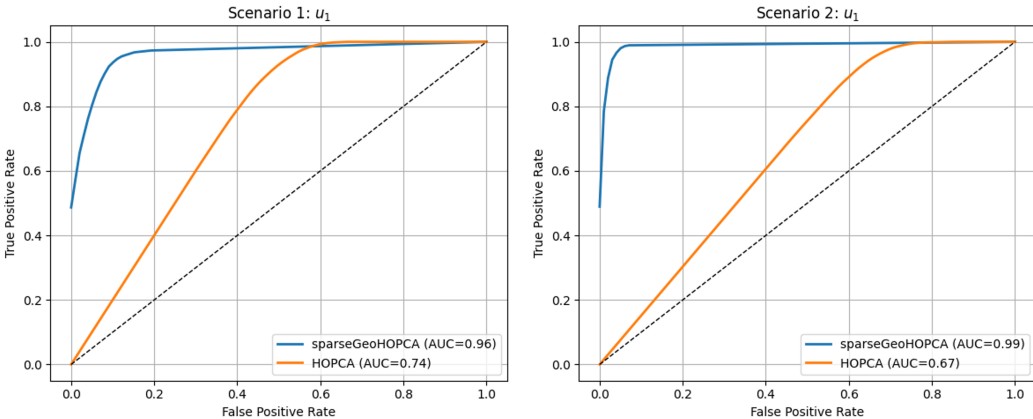

Figure 2: ROC curves for mode-$u_1$ in Scenarios 1 and 2, where it is the only sparse mode. Results are averaged over fifty replicates. Geo achieves markedly higher AUC, indicating more accurate recovery of the true support compared to HOPCA.

## 4.2 CLASSIFICATION WITH COMPRESSED FEATURES EXTRACTED

To assess the effectiveness of Geo in preserving discriminative structures under compression, we conduct classification experiments on a reduced MNIST dataset. We apply the method to the sample mode of the training tensor and compress each class-specific subspace by retaining only a small number of representative directions. The test data remain uncompressed and are projected onto these compressed bases.

Table 1 reports the overall classification accuracy under varying compression ratios. Even with a $10\times$ reduction in training dimensionality, the accuracy remains stable, demonstrating the robustness of the sparse representations learned.

Table 1: Overall classification accuracy (%) under varying compression ratios

| Compression Ratio | 1.0 (no compression) | 0.8 | 0.6 | 0.4 | 0.2 | 0.1 |
|---|---|---|---|---|---|---|
| Accuracy (%) | 87.75 | 86.25 | 86.50 | 86.88 | 86.50 | 84.62 |

Additional experimental details, including the classification framework, visualizations, and confusion matrices before and after compression, are provided in Appendix G.

In addition, we evaluate our method on the StarPlus fMRI dataset (Just, 2002). Results show that the method maintains high recognition accuracy even under substantial compression, indicating that discriminative information is preserved in real high-dimensional fMRI data as well. Full experimental details, are provided in Appendix K.

## 4.3 IMAGE RECONSTRUCTION

To further assess the applicability of Geo to high-dimensional data, we conduct an image reconstruction experiment using selected samples from the ImageNet dataset (Russakovsky et al., 2015). As most existing sparse tensor PCA methods lack public implementations and are designed for specific tensor structures, we compare our method with two representative matrix-based sparse PCA approaches.

We select two state-of-the-art sparse PCA methods designed for matrix data as baselines: sparsePCAChan (Chan et al., 2015)(Chan for short) and sparsePCABD (Del Pia et al., 2025)(BD for short). RGB images are first converted into matrix format by flattening the spatial dimensions. After extracting a fixed number of sparse components, we reconstruct the original image and compare the visual quality.

Figure 3 displays the reconstruction results. Geo produces visibly sharper edges, fewer compression artifacts, and better texture preservation across all four image examples, despite being originally designed for tensors. It also achieves consistently lower reconstruction error and reduced runtime relative to matrix-based baselines. Table 2 further confirms these observations with quantitative metrics (MSE, PSNR, SSIM), where Geo consistently outperforms Chan and BD on both Image 1 and Image 2. These findings highlight both the accuracy and efficiency of Geo, demonstrating its generalizability in structured dimensionality reduction tasks.

Full experimental details, including preprocessing steps, component settings, and descriptions of Chan and BD, as well as additional experimental results, are provided in Appendix H.

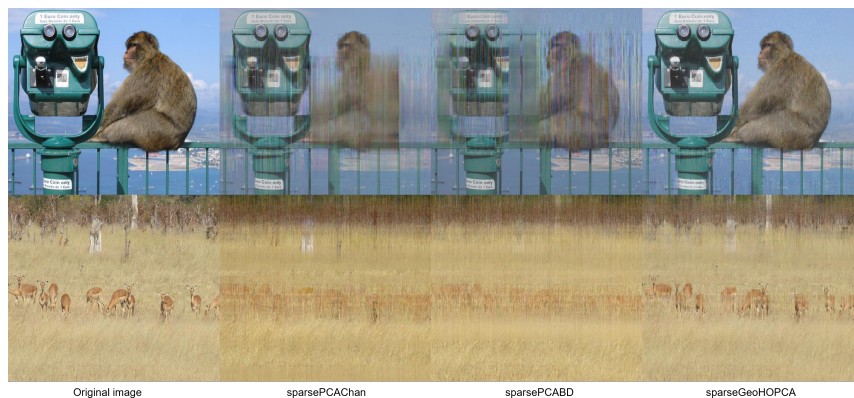

Original image          sparsePCAChan          sparsePCABD          sparseGeoHOPCA

Figure 3: Visual comparison of image reconstruction results on ImageNet samples. From left to right: original image, Chan, BD, and our Geo. From top to bottom: Image 1 and Image 2. Compared to matrix-based baselines, Geo produces reconstructions with sharper structures and fewer directional artifacts, more faithfully preserving the fine details of the original images.

Table 2: Reconstruction performance comparison on Image 1 and Image 2 in Fig. 3.

| Method | Image 1 | | | | | Image 2 | | | | |
|--------|-------|------|---------|-------|-------|-------|------|---------|-------|-------|
| | Error | Time | MSE | PSNR | SSIM | Error | Time | MSE | PSNR | SSIM |
| Chan | 70.3 | **0.8** | 0.00341 | 24.68 | 0.774 | 44.4 | 0.9 | 0.00566 | 22.47 | 0.552 |
| BD | 89.5 | 2.3 | 0.00512 | 21.90 | 0.614 | 44.2 | 2.5 | 0.00349 | 24.59 | 0.592 |
| Geo | **32.3** | **0.8** | **0.00186** | **27.32** | **0.807** | **35.5** | **0.8** | **0.00224** | **26.49** | **0.677** |

## 4.4 SEMANTIC REDUCTION AND COMPRESSION

Retrieval-Augmented Generation (RAG) (Lewis et al., 2020) and Chain-of-Thought (CoT) reasoning (Wei et al., 2022) are widely used to enhance lightweight LLMs in embodied intelligence. RAG injects factual cues (You et al., 2025), while CoT provides explicit reasoning structure (Nguyen et al., 2023), both relying heavily on semantic embeddings.

However, the high dimensionality of these embeddings has led recent works to adopt a two-stage compression pipeline: clustering to segment semantic space, followed by local dimensionality reduction. Auto-CoT (Zhang et al., 2022) and CDW-CoT (Fang et al., 2025) show that math questions form clear semantic clusters using K-Means and PCA, while xRAG (Cheng et al., 2024) and Li et al. (2025) demonstrate that document clusters can be compressed into compact representative embeddings for efficient retrieval.

To evaluate structural preservation under local dimensionality reduction, we conduct experiments on GSM8K (Cobbe et al., 2021), AQuA-RAT (Ling et al., 2017), MetaMathFewshot (Abacus.AI Team, 2024), and CSQA (Talmor et al., 2019). Following (Zhang et al., 2022), we encode all questions with `all-MiniLM-L6-v2`, cluster using K-Means (Fang et al., 2025), and compare GeoSPCA against PCA (Jolliffe, 2011), HOPCA (Nouy, 2019), and SparsePCA (Zou et al., 2006), with all methods reduced to the same output dimensionality.

Table 3: Comparison of cluster consistency and speed-up of across four datasets.

| Dataset | PCA | HOPCA | SHOPCA | GeoSPCA | Speed-up |
|---|---|---|---|---|---|
| GSM8K | 0.5866 | 0.4768 | 0.2042 | 0.1428 | $59.4\times - 541.4\times$ |
| AQuA-RAT | 0.6744 | 0.6793 | 0.3296 | 0.3033 | $30.2\times - 200.5\times$ |
| CSQA | 0.4256 | 0.4737 | 0.1363 | 0.1337 | $15.0\times - 166\times$ |
| MetaMathFewshot | 0.7714 | 0.7711 | 0.3051 | 0.1913 | $10.3\times - 85.4\times$ |

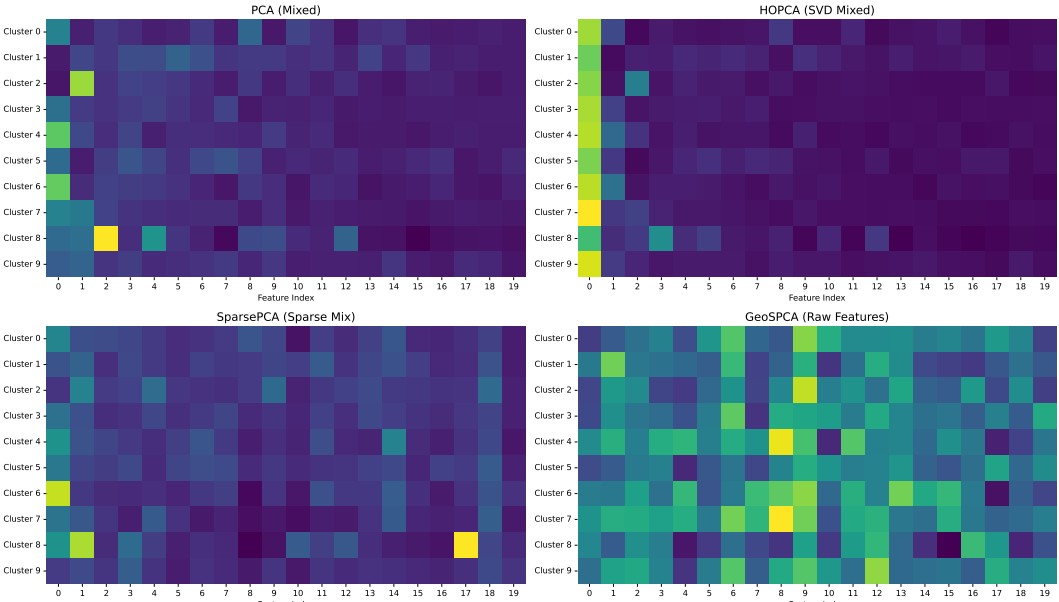

Figure 4: Comparison of semantic structure and interpretability preservation on the GSM8K dataset.

Across all datasets, GeoSPCA achieves the best memory and runtime efficiency while preserving semantic cluster structure. Heat maps (e.g., Fig. 4) show that PCA and HOPCA distort cluster boundaries, and SparsePCA partially restores sparsity but remains weakly aligned with the original semantic geometry. GeoSPCA, by directly selecting semantically meaningful embedding dimensions, yields compact and well-localised cluster–feature correspondence, maintaining geometric structure even under extreme compression. It also provides stronger reconstruction quality than SparsePCA (Table 3). Full configurations are provided in Appendix I.

## 5 CONCLUSION

We presented sparseGeoHOPCA, a geometry-aware framework for SHOPCA. By reformulating mode-wise sparse optimization as structured binary linear programs, our method eliminates the need for covariance estimation and deflation, enabling scalable and interpretable tensor decomposition.

Theoretical results establish the equivalence of the proposed subproblems to the original SHOPCA formulation and provide worst-case error bounds based on PCA residuals. The algorithm achieves linear computational complexity with respect to the tensor size via alternating optimization.

Extensive experiments confirm that sparseGeoHOPCA achieves superior support recovery, robust classification under 10× compression, and effective image reconstruction, outperforming baseline methods while preserving structural discriminability in high-dimensional settings.

## REPRODUCIBILITY STATEMENT

The Python implementation of the proposed method is available in the supplementary materials.

## ETHICS STATEMENT

This work does not involve human subjects, personally identifiable data, or sensitive information. All datasets used are publicly available and widely adopted in the research community. The proposed methods focus on algorithmic development and empirical validation. We believe this work raises no ethical concerns in relation to the ICLR Code of Ethics.

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

# A   PROOF OF THEOREM 3.1

This appendix provides the detailed proof of Theorem 3.1. Let $\{U_i\}_{i=1,\,i\neq n}^N$ denote the collection of all mode-$i$ projection matrices excluding mode $n$, i.e., $(U_1, \ldots, U_{n-1}, U_{n+1}, \ldots, U_N)$. We denote the objective function in 3 as $f(U_n)$ when all other $U_i$'s are fixed. Assume that each $U_i$ is column-orthonormal.

By properties of mode-wise tensor projections and the orthogonality of $U_i U_i^\top$, we have:

$$
\begin{aligned}
f(U_n) &= \|\mathcal{X} - \mathcal{X} \times_1 U_1 U_1^\top \cdots \times_N U_N U_N^\top\|_F^2 \\
&= \|\mathcal{X} \times_N U_N U_N^\top - \mathcal{X} \times_1 U_1 U_1^\top \cdots \times_N U_N U_N^\top\|_F^2 \\
&\quad + \|\mathcal{X} - \mathcal{X} \times_N U_N U_N^\top\|_F^2 \\
&\;\;\vdots \\
&= \|\mathcal{X} \times_{j\neq n} U_j U_j^\top - \mathcal{X} \times_{j\neq n} U_j U_j^\top \times_n U_n U_n^\top\|_F^2 \\
&\quad + \|\mathcal{X} - \mathcal{X} \times_{j\neq n} U_j U_j^\top\|_F^2,
\end{aligned} \tag{9}
$$

where $\times_{j\neq n}$ denotes the sequence of mode-$j$ projections over all $j \in \{1, \ldots, N\} \setminus \{n\}$. Noting that the second term in the final expression is independent of $U_n$, the optimization reduces to minimizing $\|\mathcal{X} \times_{j\neq n} U_j^\top - \mathcal{X} \times_{j\neq n} U_j^\top \times_n U_n U_n^\top\|_F^2$. Note that under the setting where the factor matrix $U_n$ is updated iteratively. If $U_n$ is instead generated in a single step, or when no initialization is provided, then $\mathrm{unfold}_n(\mathcal{X} \times_{j\neq n} U_j U_j^\top)$ should be directly taken as $X_{(n)}$. This completes the proof.

# B   PROOF OF THEOREM 3.2

This appendix provides a detailed proof of Theorem 3.2. The proof is divided into two main steps.

**Step 1: Reformulation of the Problem.**   We first show that the optimization problem 5 can be equivalently reformulated as the following problem:

$$
\begin{aligned}
\underset{\mathbf{s}\in\{0,1\}^{\prod_{i\neq n} J_i},\, U_n\in\mathbb{R}^{J_n\times R_n}}{\text{maximize}} \quad & \sum_{j=1}^{\prod_{i\neq n} J_i} s_j \sum_{k=1}^{R_n} (X_{(n)}(:,j)^\top U_n(:,k))^2 \\
\text{subject to} \quad & \mathbf{e}^\top \mathbf{s} \leq k_n, \quad U_n^\top U_n = I_{R_n}.
\end{aligned} \tag{10}
$$

*Proof of Step 1.*

By the property of the Frobenius norm, problem 5 can be rewritten as:

$$
\begin{aligned}
\underset{U_n}{\text{minimize}} \quad \|X_{(n)} - U_n U_n^\top X_{(n)}\|_F^2 &= \mathrm{tr}((X_{(n)} - U_n U_n^\top X_{(n)})^\top (X_{(n)} - U_n U_n^\top X_{(n)})) \\
&= \mathrm{tr}(X_{(n)}^\top X_{(n)}) - \mathrm{tr}(X_{(n)}^\top U_n U_n^\top X_{(n)}),
\end{aligned} \tag{11}
$$

where the second equality follows from the orthogonality condition $U_n^\top U_n = I_{R_n}$.

Since the first term $\mathrm{tr}(X_{(n)}^\top X_{(n)})$ is constant with respect to $U_n$, minimizing the objective is equivalent to maximizing

$$
\underset{U_n}{\text{maximize}} \quad \mathrm{tr}(X_{(n)}^\top U_n U_n^\top X_{(n)}) = \underset{U_n}{\text{maximize}} \quad \mathrm{tr}(U_n^\top X_{(n)} X_{(n)}^\top U_n). \tag{12}
$$

The optimization problem 12 is originally formulated as a left eigenvalue problem. We first transform it into an equivalent right eigenvalue formulation to facilitate the incorporation of sparsity constraints.

Specifically, we introduce an auxiliary binary vector $\mathbf{s} \in \{0,1\}^{\prod_{i\neq n} J_i}$, where $s_i = 0$ if the $i$-th row of $W$ is zero, and $s_i = 1$ otherwise. Define $S = \mathrm{diag}(\mathbf{s})$. The constraint $\mathbf{e}^\top \mathbf{s} \leq k_n$ ensures that the number of nonzero rows of $W$ does not exceed $k_n$.

Under this construction, the optimization problem becomes

$$
\underset{\mathbf{s}\in\{0,1\}^{\Pi_{i\neq n} J_i},\ W\in\mathbb{R}^{\Pi_{i\neq n} J_i \times R_n}}{\text{maximize}} \quad \text{tr}(W^\top S X_{(n)}^\top X_{(n)} S W)
$$
$$
\text{subject to} \quad \mathbf{e}^\top \mathbf{s} \leq k_n, \quad W^\top W = I_{R_n}, \quad W^\top S W = I_{R_n}. \tag{13}
$$

The additional constraint $W^\top S W = I_{R_n}$ can be removed without loss of optimality, as it can be justified through singular value decomposition (SVD) analysis of $S X_{(n)}^\top = X_{(n)}(:, s)$.

We then convert the right eigenvalue formulation back to the original left eigenvalue form. By expanding the trace, the objective simplifies as:

$$
\text{tr}(U_n^\top X_{(n)} X_{(n)}^\top U_n) = \sum_{j=1}^{k_n} \sum_{k=1}^{R_n} (X_{(n)}(:, s_j)^\top U_n(:, k))^2
$$
$$
= \sum_{j=1}^{\Pi_{i\neq n} J_i} s_j \sum_{k=1}^{R_n} (X_{(n)}(:, j)^\top U_n(:, k))^2, \tag{14}
$$

thus completing the proof that problem 5 is equivalent to problem 10.

**Step 2: Connection to the $\eta$-Constrained Selection Problem.** Let $\mathbf{s}^0$ be an optimal solution to problem 10. We now show that there exists a constant $\delta > 0$ such that for any $\eta \in [\eta(\mathbf{s}^0), \eta(\mathbf{s}^0) + \delta]$, any optimal solution to the $\eta$-constrained selection problem 6 is also an optimal solution to problem 10.

*Proof of Step 2.*

We proceed by contradiction. Suppose $\eta = \eta(\mathbf{s}^0)$, and assume that there exists a selection $\mathbf{s}' \in \{0,1\}^{\Pi_{i\neq n} J_i}$ such that $\mathbf{s}'$ is an optimal solution to problem 6, but not an optimal solution to problem 10.

Then it follows that

$$
\sum_{j=1}^{\Pi_{i\neq n} J_i} s_j^0 \sum_{k=1}^{R_n} (X_{(n)}(:, j)^\top U_n(:, k))^2 > \sum_{j=1}^{\Pi_{i\neq n} J_i} s_j' \sum_{k=1}^{R_n} (X_{(n)}(:, j)^\top U_n(:, k))^2. \tag{15}
$$

Since $\mathbf{s}'$ is feasible for problem 6, we have

$$
\sum_{j=1}^{\Pi_{i\neq n} J_i} s_j' \|X_{(n)}(:, j)\|_F^2 \geq \sum_{j=1}^{\Pi_{i\neq n} J_i} s_j^0 \|X_{(n)}(:, j)\|_F^2. \tag{16}
$$

Combining 15 and 16, we conclude that

$$
\eta(\mathbf{s}') > \eta(\mathbf{s}^0) = \eta,
$$

which contradicts the feasibility condition $\eta(\mathbf{s}) \leq \eta$ required in problem 6.

Since the feasible set $\{0,1\}^{\Pi_{i\neq n} J_i}$ is finite and discrete, there exists a constant $\delta > 0$ such that the set of feasible solutions to problem 6 remains unchanged when $\eta$ varies within $[\eta(\mathbf{s}^0), \eta(\mathbf{s}^0) + \delta]$.

Specifically, let $\hat{\mathbf{s}}$ be defined as

$$
\hat{\mathbf{s}} \in \arg\min_{\mathbf{s}} \{\|X_{(n)}(:, \mathbf{s}) - U_n[\mathbf{s}] U_n[\mathbf{s}]^\top X_{(n)}(:, \mathbf{s})\|_F^2 > \eta(\mathbf{s}^0)\}. \tag{17}
$$

and define $\hat{\eta} = \|X_{(n)}(:, \hat{\mathbf{s}}) - U_n[\hat{\mathbf{s}}] U_n[\hat{\mathbf{s}}]^\top X_{(n)}(:, \hat{\mathbf{s}})\|_F^2$. Then, we can set $\delta = \frac{\hat{\eta} - \eta(\mathbf{s}^0)}{2}$ to guarantee the stability of the optimal solution. This completes the proof.

Combining Step 1 and Step 2, this completes the proof of Theorem 3.2.

## C    PROOF OF THEOREM 3.3

This appendix provides the proof of Theorem 3.3. The second inequality in 7 is straightforward. We now focus on proving the first inequality.

Consider

$$
\begin{aligned}
\|S^0 \epsilon^n\|_F &= \|S^0 (X_{(n)} - VV^\top X_{(n)})\|_F \\
&= \|S^0 X_{(n)} - S^0 VV^\top X_{(n)}\|_F \\
&= \|S^0 X_{(n)} - VV^\top S^0 X_{(n)}\|_F \\
&\geq \|X_{(n)}(:, \mathbf{s}^0) - U_n[\mathbf{s}^0] U_n[\mathbf{s}^0]^\top X_{(n)}(:, \mathbf{s}^0)\|_F \\
&= \sqrt{\eta(\mathbf{s}^0)},
\end{aligned}
\tag{18}
$$

where the third equality uses $S^0 VV^\top = VV^\top S^0$, and the inequality follows from restricting the operation to the selected columns corresponding to $\mathbf{s}^0$.

This completes the proof of the boundary estimation 7.

Furthermore, when $\eta = \|S^0 \epsilon^n\|_F$, since $\|X_{(n)}(:, \mathbf{s}^0) - U_n[\mathbf{s}^0] U_n[\mathbf{s}^0]^\top X_{(n)}(:, \mathbf{s}^0)\|_F \leq \|S^0 \epsilon^n\|_F$, it follows that $\eta(\mathbf{s}^0) \leq \eta$, ensuring that $\mathbf{s}^0$ is feasible for problem 6.

## D    PROOF OF THEOREM 3.4

This appendix provides the proof of Theorem 3.4. We have

$$
\begin{aligned}
f(U_1, U_2, \ldots, U_N) &= \|\mathcal{X} - \mathcal{X} \times_1 U_1 U_1^\top \times_2 \cdots \times_N U_N U_N^\top\|_F^2 \\
&\leq \sum_{n=1}^N \|\mathcal{X} - \mathcal{X} \times_n U_n U_n^\top\|_F^2 \\
&= \sum_{n=1}^N \|X_{(n)} - U_n U_n^\top X_{(n)}\|_F^2 \\
&\leq \sum_{n=1}^N \sum_{i \in \sigma^n} \|\epsilon_i^n\|^2,
\end{aligned}
\tag{19}
$$

where the first inequality follows from the results in Kolda & Bader (2009); Hong et al. (2020), and the second inequality follows from Theorem 3.3.

## E    DETAILS OF COMPLEXITY ANALYSIS

This appendix provides additional details supporting the complexity analysis presented in Section 3.4. In particular, we explain (i) why the number of cuts generated by the cutting-plane procedure may be exponential in the worst case, and (ii) why the binary linear optimization (BLO) subproblem solved at each iteration nevertheless admits an $O(k_n)$-time solution.

### E.1    WORST-CASE NUMBER OF CUTS

Following the construction in Bertsimas & Kitane (2022), one can explicitly design a matrix $X \in \mathbb{R}^{p \times q}$ in mode $n$ for which the cutting-plane algorithm must eliminate every binary support except for the single optimal one. We briefly summarize the argument here.

Consider the case $k_n = 2$ and $p = q$, and let the columns of $X$ be constructed so that

- the first two columns satisfy $\|X(:, 1)\|_2 = \|X(:, 2)\|_2 = 1$,
- all remaining columns satisfy $\|X(:, j)\|_2 > 1$ for $j \geq 3$, and
- for every pair $(i, j) \neq (1, 2)$ or $(2, 1)$ there exists a unit vector $u \in \mathbb{R}^q$ with $u^\top X(:, i, j)^\top X(:, i, j) u < 2$, whereas the pair $(1, 2)$ satisfies $u^\top X(:, 1, 2)^\top X(:, 1, 2) u = 2$.

Here $X(:,i,j)$ denotes the $p \times 2$ matrix consisting of columns $i$ and $j$ of $X$. Under this construction, only the binary vector selecting columns $(1,2)$ yields an objective value exceeding a prescribed threshold $\eta$, while every other binary vector violates the associated inequality.

Since each generated cut removes exactly one infeasible binary vector by setting $\eta = 0.000001$ for example, the algorithm must add one cut for every non-optimal support. The total number of possible supports is $\binom{q}{k_n} = \binom{q}{2}$, which grows quadratically in $p$ for $k_n = 2$ and exponentially in general. Hence, in the worst case, the cutting-plane method may indeed require an exponential number of iterations before converging to the optimal binary solution.

### E.2 COMPLEXITY OF THE BLO SUBPROBLEM

Although the number of cuts may be exponential, the cost of solving the BLO subproblem at each iteration remains low. As shown in Bertsimas & Kitane (2022), the linear objective defining the BLO in mode $n$ is separable across coordinates and depends only on the sum of the $k_n$ largest entries of a score vector derived from the separation oracle. Let $w \in \mathbb{R}^{\prod_{i \neq n} J_i}$ denote this score vector. The BLO problem at iteration $t$ can therefore be written as

$$\max_{s \in \{0,1\}^{\prod_{i \neq n} J_i}} \langle w, s \rangle \qquad \text{s.t. } \|s\|_0 = k_n, \ s \neq s^{(1)}, \ldots, s^{(t-1)}, \tag{20}$$

where $s^{(1)}, \ldots, s^{(t-1)}$ are the supports eliminated in earlier iterations.

Because the objective is additive and the constraint enforces exactly $k_n$ active components, the optimal solution corresponds to selecting the indices of the $k_n$ largest entries of $w$, except that previously eliminated supports are skipped. Enumerating these supports can be organized as a depth-$k_n$ tree search in which each level selects the next largest remaining coordinate. The branching factor decreases monotonically as the search descends the tree, and the total number of visited nodes is $O(k_n)$ in the worst case.

Therefore, as established in Bertsimas & Kitane (2022), the BLO subproblem can be solved in $O(k_n)$ time, and the per-iteration complexity of the overall algorithm is dominated by the $O(k_n^3 + J_n k_n^2)$ cost of the SVD required by the separation oracle.

This completes the detailed justification for the complexity bounds used in the main text.

## F ADDITIONAL DETAILS ON SYNTHETIC EXPERIMENTS

### F.1 SIMULATION SETUP

We consider four simulation scenarios that vary in dimensionality and sparsity structure:

- Scenario 1: $100 \times 100 \times 100$, sparsity in mode $\mathbf{U}$ only;
- Scenario 2: $1000 \times 20 \times 20$, sparsity in mode $\mathbf{U}$ only;
- Scenario 3: $100 \times 100 \times 100$, sparsity in all three modes;
- Scenario 4: $1000 \times 20 \times 20$, sparsity in all three modes.

For sparse modes, we randomly set 50% of entries to zero, and the remaining entries are drawn from $N(0,1)$. For dense modes, the factors are obtained as the first $K$ left and right singular vectors of matrices with i.i.d. $N(0,1)$ entries.

### F.2 ROC ANALYSIS AND FEATURE SELECTION ACCURACY

To quantify the accuracy of support recovery, we report averaged Receiver Operating Characteristic (ROC) curves over 50 replications in each simulation setting. Figure 2 displays ROC curves for mode-$u_1$ in Scenarios 1 and 2, where it is the only sparse mode. Figure 5 presents ROC curves for modes $u_1$, $v_1$, and $w_1$ in Scenarios 3 and 4, where sparsity is present in all modes. In both cases, we compare the proposed *sparseGeoHOPCA* method with a baseline HOPCA approach that applies naive thresholding to the components of Tucker decomposition (Kolda & Bader, 2009; Kossaifi et al., 2019).

As illustrated in Figures 2 and 5, *sparseGeoHOPCA* consistently achieves higher true positive rates while maintaining substantially lower false positive rates across all simulation settings. This improvement is particularly evident in Scenarios 3 and 4, where the data exhibit full-mode sparsity and severe dimensional imbalance. Moreover, the area under the ROC curve (AUC) highlights the robustness and reliability of sparseGeoHOPCA in high-dimensional, sparse tensor settings.

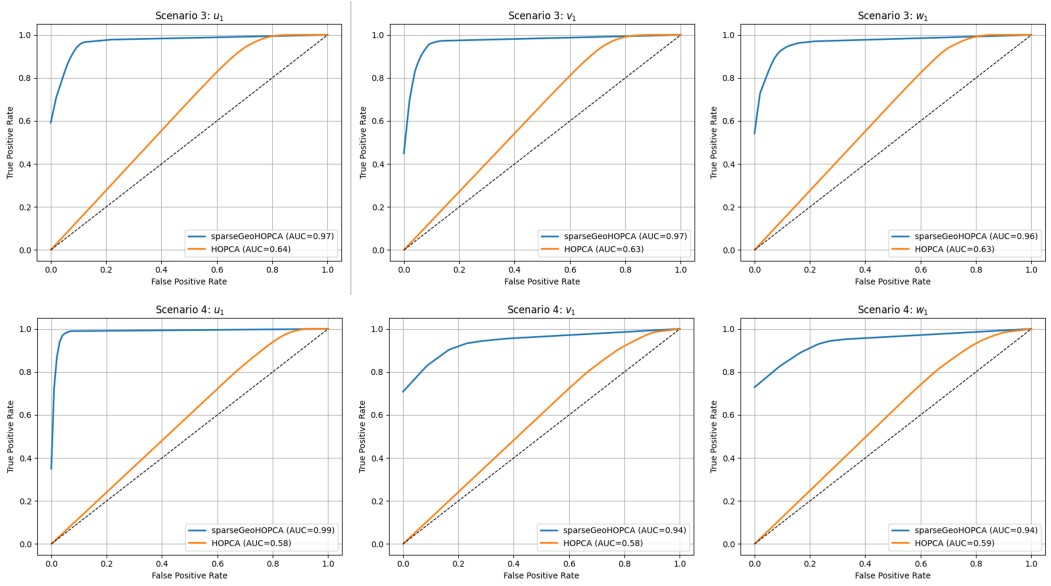

Figure 5: ROC curves for modes $u_1$, $v_1$, and $w_1$ in Scenarios 3 and 4, where sparsity is present in all modes. Results are averaged over fifty independent runs.

Table 4: TP/FP comparison between HOPCA and sparseGeoHOPCA under Scenario 3 with different noise types.

| Noise Type | Mode | Method | TP (mean ± std) | FP (mean ± std) |
|---|---|---|---|---|
| Gaussian Noise | $u_1$ | sparseGeoHOPCA | 0.957 ± 0.054 | 0.028 ± 0.041 |
| | | HOPCA | 1.000 ± 0.000 | 0.725 ± 0.073 |
| | $v_1$ | sparseGeoHOPCA | 0.977 ± 0.035 | 0.033 ± 0.044 |
| | | HOPCA | 1.000 ± 0.000 | 0.739 ± 0.081 |
| | $w_1$ | sparseGeoHOPCA | 0.973 ± 0.041 | 0.032 ± 0.050 |
| | | HOPCA | 1.000 ± 0.000 | 0.727 ± 0.081 |
| Mixed Noise | $u_1$ | sparseGeoHOPCA | 0.960 ± 0.050 | 0.030 ± 0.052 |
| | | HOPCA | 1.000 ± 0.000 | 0.770 ± 0.079 |
| | $v_1$ | sparseGeoHOPCA | 0.972 ± 0.045 | 0.030 ± 0.042 |
| | | HOPCA | 1.000 ± 0.000 | 0.787 ± 0.063 |
| | $w_1$ | sparseGeoHOPCA | 0.956 ± 0.048 | 0.026 ± 0.044 |
| | | HOPCA | 1.000 ± 0.000 | 0.796 ± 0.077 |

The TP/FP comparison in Table 4 shows that under both Gaussian noise and mixed noise, *sparseGeoHOPCA* achieves consistently high true-positive rates (approximately 0.95–0.98) while keeping false positives extremely low (around 0.03). In contrast, HOPCA attains perfect TP values but suffers from a large number of false activations, with FP ranging from 0.72 to 0.80 across all modes and both noise types.

For mixed noise, we adopt a Mixture-of-Gaussians (MoG) formulation following the noise modeling strategy introduced in Chen et al. (2018), where the observed tensor is perturbed by a combination of Gaussian components with different variances to simulate heterogeneous and heavy-tailed noise patterns. This setting captures more realistic distortions than single Gaussian noise, making the support-recovery task substantially more challenging.

Even under such complex noise, *sparseGeoHOPCA* remains highly robust: it accurately recovers the true support while effectively suppressing noise-driven spurious entries. Overall, the table highlights that *sparseGeoHOPCA* provides strong noise robustness and precise sparsity control compared with covariance-based HOPCA.

### F.3 TRUE AND FALSE POSITIVE RATE COMPARISON

Table 5 summarizes the mean and standard deviation of true positive (TP) and false positive (FP) rates for sparseGeoHOPCA and HOPCA across all modes and scenarios. The results reinforce the ROC analysis by demonstrating that sparseGeoHOPCA not only yields high TP rates but also significantly reduces FP rates compared to HOPCA.

Table 5: TP/FP comparison between HOPCA and sparseGeoHOPCA across scenarios

| Scenario | Mode | Method | TP (mean ± std) | FP (mean ± std) |
|----------|------|--------|-----------------|-----------------|
| 1 | $u_1$ | sparseGeoHOPCA | $0.967 \pm 0.048$ | $0.041 \pm 0.052$ |
|   |       | HOPCA | $1.000 \pm 0.000$ | $0.514 \pm 0.082$ |
| 2 | $u_1$ | sparseGeoHOPCA | $0.988 \pm 0.015$ | $0.012 \pm 0.017$ |
|   |       | HOPCA | $1.000 \pm 0.000$ | $0.670 \pm 0.077$ |
| 3 | $u_1$ | sparseGeoHOPCA | $0.972 \pm 0.040$ | $0.034 \pm 0.051$ |
|   |       | HOPCA | $1.000 \pm 0.000$ | $0.730 \pm 0.076$ |
|   | $v_1$ | sparseGeoHOPCA | $0.968 \pm 0.048$ | $0.031 \pm 0.038$ |
|   |       | HOPCA | $1.000 \pm 0.000$ | $0.745 \pm 0.071$ |
|   | $w_1$ | sparseGeoHOPCA | $0.962 \pm 0.054$ | $0.034 \pm 0.053$ |
|   |       | HOPCA | $1.000 \pm 0.000$ | $0.733 \pm 0.076$ |
| 4 | $u_1$ | sparseGeoHOPCA | $0.989 \pm 0.016$ | $0.014 \pm 0.017$ |
|   |       | HOPCA | $1.000 \pm 0.000$ | $0.839 \pm 0.072$ |
|   | $v_1$ | sparseGeoHOPCA | $0.928 \pm 0.098$ | $0.044 \pm 0.091$ |
|   |       | HOPCA | $1.000 \pm 0.000$ | $0.845 \pm 0.112$ |
|   | $w_1$ | sparseGeoHOPCA | $0.929 \pm 0.096$ | $0.045 \pm 0.095$ |
|   |       | HOPCA | $1.000 \pm 0.000$ | $0.826 \pm 0.120$ |

While HOPCA achieves perfect TP rates in all cases, it suffers from excessive false positives, often exceeding 70% in more challenging configurations, indicating poor feature specificity. In contrast, sparseGeoHOPCA delivers a more balanced and controlled feature selection. Notably, in Scenario 4 mode $v_1$, sparseGeoHOPCA achieves a TP rate of $0.928 \pm 0.098$ and an FP rate of $0.044 \pm 0.091$, whereas HOPCA yields an FP rate as high as $0.845 \pm 0.112$.

These findings demonstrate that sparseGeoHOPCA provides a more effective solution for sparse tensor decomposition when accurate support recovery is critical, especially under limited sample sizes and high ambient dimensionality.

## G CLASSIFICATION WITH COMPRESSED FEATURES EXTRACTED

The proposed sparseGeoHOPCA algorithm is designed to extract informative and sparse representations from high-dimensional tensor data while preserving structural integrity across all modes. It enables simultaneous multimodal co-clustering and feature selection. To validate the effectiveness of the method in real-world classification tasks, we apply it to handwritten digit recognition using the MNIST dataset (LeCun & Cortes, 1998). MNIST is a benchmark dataset comprising 60,000 training and 10,000 test images, where each image is a $28 \times 28$ grayscale representation of a digit (0–9).

The fundamental assumptions underlying image classification are: (1) samples from the same class share common latent features, and (2) samples from different classes exhibit distinct structural patterns. Therefore, a successful classification algorithm should be able to extract discriminative features from the raw data. In this study, we focus on supervised classification, where class labels are known during training.

We adopt a projection-based classification framework, following the methodology proposed in Keegan et al. (2021); Newman et al. (2017). In the training phase, a subspace of local features (or "local basis") is constructed for each class using the respective training samples. This subspace captures intra-class variation. During testing, each test image $b$ is orthogonally projected onto all subspaces specific to the class, and the classification decision is made by selecting the subspace that yields the smallest Frobenius norm between the projection and the original sample. Figure 6 illustrates this

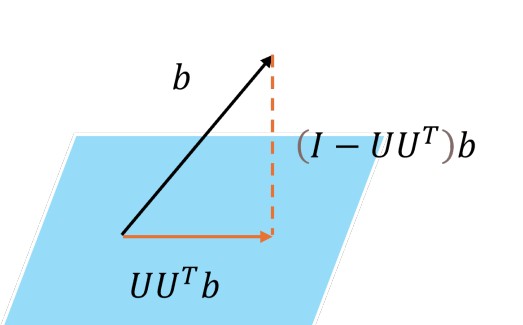

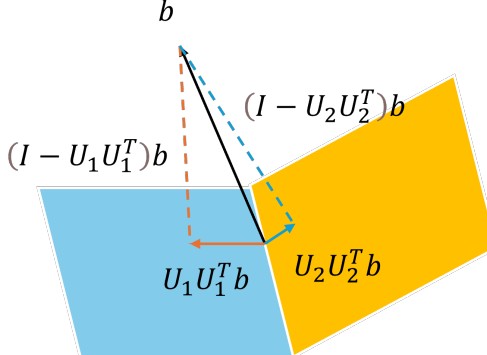

(a) Projection of a test image $b$ onto a single subspace $UU^T$. The residual is $(I - UU^T)b$.

(b) Projection of $b$ onto multiple class-specific subspaces $U_1U_1^T$, $U_2U_2^T$, etc.

Figure 6: Illustration of the projection-based classification scheme. Classification is based on the subspace yielding the smallest projection residual.

framework. In the left panel, a test image $b$ is projected onto a single class-specific subspace $UU^T$, with residual $(I - UU^T)b$. In practice, the image is projected onto multiple subspaces $\{U_iU_i^T\}$, and the label is predicted based on the minimum projection error, as shown in the right panel.

The success of this approach critically depends on the construction of representative subspaces for each class. To this end, we apply sparseGeoHOPCA independently to the training samples of each class. This process yields sparse, interpretable basis components that enhance both intra-class consistency and inter-class separability.

To visually demonstrate the effectiveness of the extracted features, we use digits 7 and 8 from MNIST as an illustrative case. Figure 7 presents representative training images for both classes. Although intra-class variability exists, structural differences between classes are visually apparent.

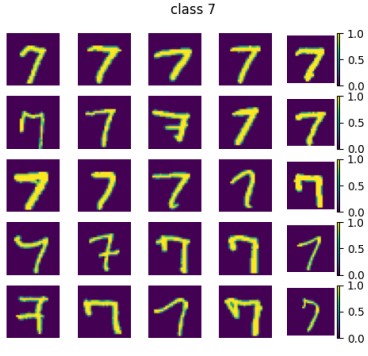

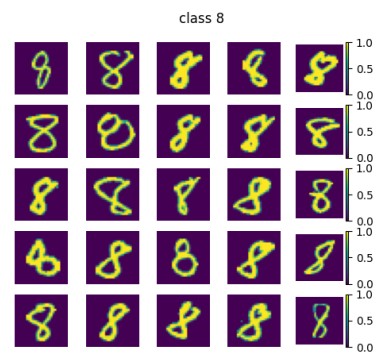

(a) Sample training images from digit class 7.

(b) Sample training images from digit class 8.

Figure 7: Training samples used to construct class-specific subspaces.

We then apply sparseGeoHOPCA to the training tensors of digits 7 and 8 and visualize the first two basis vectors extracted along the sample mode (i.e., the mode corresponding to training indices). As shown in Figure 8, the learned components capture digit-specific structures: class 7 shows strong

vertical and angular strokes, while class 8 reveals looped patterns. These localized and interpretable features improve downstream classification performance.

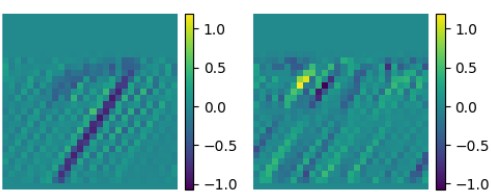
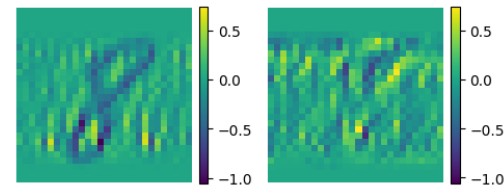

(a) First two basis elements from the sample mode for class 7.

(b) First two basis elements from the sample mode for class 8.

Figure 8: Visualization of class-specific basis vectors extracted by sparseGeoHOPCA.

**Evaluation of Classification Performance Before and After Compression**   To evaluate the effectiveness of sparseGeoHOPCA in preserving discriminative structures under compression, we conduct experiments on a reduced version of MNIST. Specifically, we use 500 training and 80 test samples per class, resulting in a total of 5,000 training and 800 test samples.

To simulate data compression, we apply sparseGeoHOPCA to the sample mode of the training tensor and reduce its dimensionality by a factor of ten. That is, for each class, only 50 representative directions are retained after decomposition. The test set remains uncompressed and is projected onto the compressed class-specific bases.

Figure 9 displays the normalized confusion matrices obtained before and after compression. The baseline case (left panel) achieves an overall accuracy of 87.75%, while the compressed case (right panel) achieves 84.62%. Despite the tenfold reduction in the training dimension, the classification structure remains largely intact. In particular, digits such as 0, 1, and 4 retain high precision, while digits such as 3, 5, and 8 are more sensitive to the compression due to higher intra-class variability.

To further analyze classification robustness under different compression levels, we evaluate the overall accuracy for varying compression ratios. As shown in Table 1, the model maintains a relatively stable accuracy even when only 10% of the training data (in the sample mode) are retained.

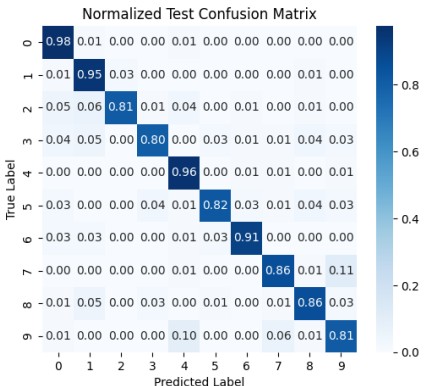
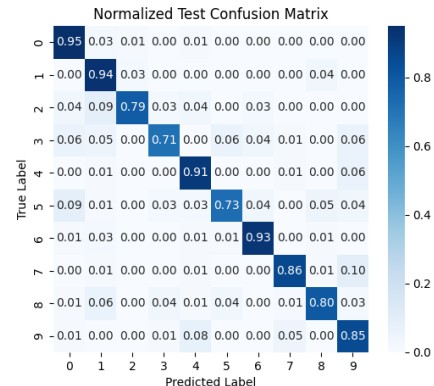

(a) Normalized confusion matrix using original data (Accuracy: 87.75%).

(b) Normalized confusion matrix after $10\times$ compression (Accuracy: 84.62%).

Figure 9: Comparison of classification performance before and after sparseGeoHOPCA-based sample-mode compression. Each class uses 500 training and 80 test samples.

## H  DETAILS OF IMAGE RECONSTRUCTION EXPERIMENT

**Dataset and Preprocessing.** We randomly select four RGB images from the ImageNet dataset (Russakovsky et al., 2015), with original resolutions of $500 \times 368 \times 3$, $500 \times 375 \times 3$, $500 \times 375 \times 3$, and $500 \times 359 \times 3$, respectively. To facilitate matrix-based analysis, each image $\mathbf{I} \in \mathbb{R}^{m \times n \times 3}$ is reshaped in two modes: (i) as a row-wise matrix $\mathbf{I}_r \in \mathbb{R}^{m \times 3n}$ by flattening the RGB channels along columns; (ii) as a column-wise matrix $\mathbf{I}_c \in \mathbb{R}^{n \times 3m}$ by flattening the RGB channels along rows. These representations allow structured feature extraction along spatial or chromatic dimensions.

**Baselines.** We compare our method against two representative state-of-the-art baselines: **(1) Chan's algorithm (sparsePCAChan):** a polynomial-time approximation algorithm with provable multiplicative guarantees for sparse PCA (Chan et al., 2015). **(2) Block-diagonalization-based method (sparsePCABD):** a heuristic approach that transforms the input matrix into a block-diagonal form to facilitate sparse component extraction (Del Pia et al., 2025).

All methods retain an equal number of principal components (90), and the reconstruction is performed by means of a linear combination of the selected bases. The visual results are shown in Figure 3 in the main text and Figure 10.

**Observations.** Compared to the matrix-specific methods, *sparseGeoHOPCA* yields sharper reconstructions with fewer vertical or horizontal artifacts. This indicates that the geometry-aware support selection mechanism, originally designed for tensors, transfers well to structured matrices, especially when preserving global structure is critical.

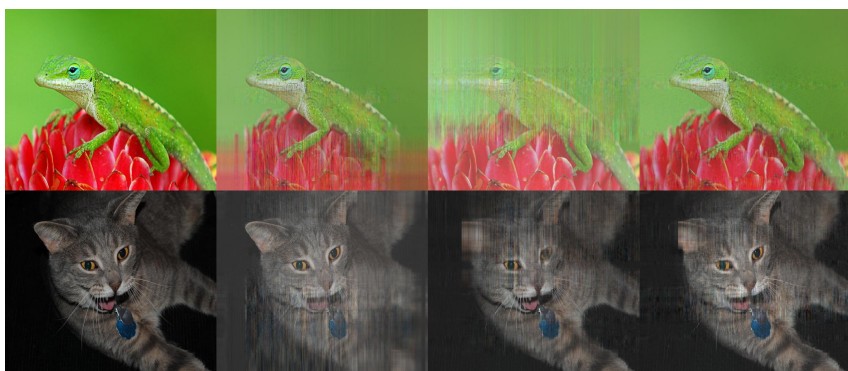

Figure 10: Visual comparison of image reconstruction results on ImageNet samples. From left to right: original image, sparsePCAChan, sparsePCABD, and our *sparseGeoHOPCA*.

Quantitative metrics (PSNR, SSIM, and MSE) are included to complement the visual findings. Compared to matrix-specific methods, *sparseGeoHOPCA* produces sharper reconstructions with fewer directional artifacts (e.g., vertical or horizontal banding). Beyond qualitative improvements, the method consistently yields lower reconstruction error (Frobenius norm) and competitive runtime, while also achieving the highest PSNR and SSIM across all test images. As shown in Table 6, *sparseGeoHOPCA* attains the lowest Frobenius error, the fastest execution in most cases, and superior perceptual quality measures. These results highlight the benefits of geometry-aware support selection, demonstrating both numerical accuracy and perceptual fidelity, even when applied to flattened matrix data.

These results suggest that the geometry-aware support selection mechanism, originally designed for tensors, transfers effectively to structured matrices, particularly when global structural fidelity is essential. Quantitative image quality metrics such as PSNR or SSIM may be incorporated into future work to complement these findings.

Table 6: Reconstruction performance comparison on Image 3 and Image 4 in Fig. 10.

| Method | Image 3 | | | | | Image 4 | | | | |
|--------|-------|------|---------|-------|-------|-------|------|---------|-------|-------|
| | Error | Time | MSE | PSNR | SSIM | Error | Time | MSE | PSNR | SSIM |
| Chan | 67.6 | **0.8** | 0.00814 | 20.89 | 0.764 | 39.5 | 0.8 | 0.01279 | 18.93 | 0.393 |
| BD | 53.6 | 2.4 | 0.00227 | 26.46 | 0.788 | 32.4 | 2.1 | 0.00964 | 20.16 | 0.470 |
| Geo | **33.5** | **0.8** | **0.00204** | **26.92** | **0.811** | **30.9** | **0.6** | **0.00177** | **27.52** | **0.765** |

# I    SEMANTIC DEGRADATION AND COMPRESSION

**Datasets.** We conduct experiments on four representative datasets commonly evaluated in CoT and RAG applications, which cover arithmetic reasoning, symbolic manipulation, commonsense inference and large-scale mathematical problem solving. To evaluate the robustness and scalability of our work, we adopt datasets that span different orders of magnitude from medium-sized to large-scale datasets. Specifically, we setup and use: *1) CommonsenseQA* (Talmor et al., 2019): 1,000 commonsense multiple-choice questions (100 validated sets). *2) GSM8K* (Cobbe et al., 2021): 5,,000 grade-school arithmetic problems (500 validated sets). *3) AQUA-RAT* (Ling et al., 2017): 10,000 multiple-choice arithmetic reasoning questions (1,000 validated sets). *4) MetaMathFewshot* (Abacus.AI Team, 2024): 40,000 augmented mathematical reasoning problems (4,000 validated sets).

**Experimental Design.** All questions are encoded into dense 384-dimensional embeddings using the `all-MiniLM-L6-v2` sentence transformer by referring to scenario (Zhang et al., 2022). To establish a structural ground-truth baseline, we apply K-Means clustering with $K = 10$ to the entire embedding space as in (Fang et al., 2025). A validated set was also introduced to evaluate efficiency performance. We compare our method with three representative methods based on sparsity and covariance: PCA (Jolliffe, 2011), HOPCA (Nouy, 2019) and SparsePCA (Zou et al., 2006). After clustering, all methods reduce the embeddings to a unified dimensionality of $K' = 20$ for equitable comparison. The evaluation focuses on semantic dimensionality reduction and compression of the question embeddings in each cluster in: *1) Processing Efficiency*: measured as the latency of dimensionality reduction in semantic clusters. *2) Cluster Consistency*: quantified by adjusted rand index (ARI) against full-dimensional clustering. *3) Semantic Maintenance and Explainability*: assessed through cluster–feature heatmaps visualising average absolute feature activation within each cluster.

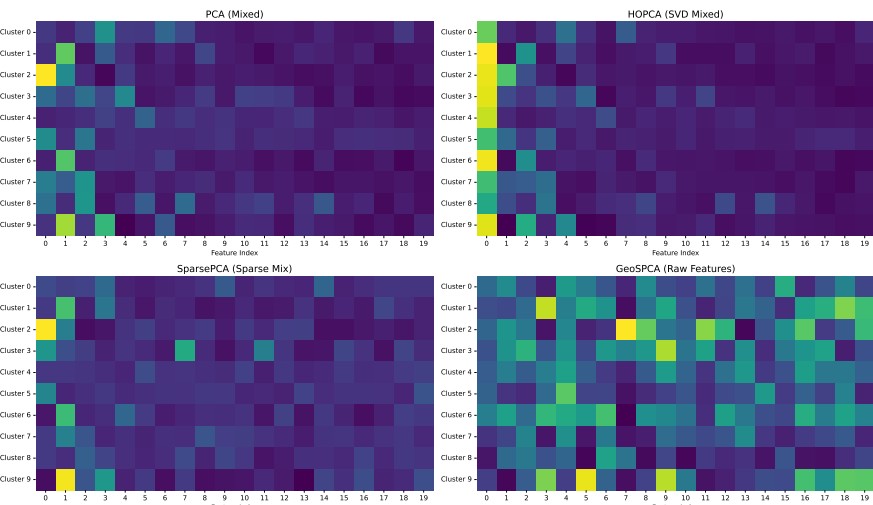

Figure 11: Comparison of semantic structure and interpretability reservation on AQuA.

**Observations.** In semantic dimension reduction and compression tasks of semantic clusters for CoT and RAG, the main concerns are the preservation of semantic interpretability, the computational efficiency of projection and the reconstruction fidelity relative to the original embedding space. Across all four datasets, GeoSPCA consistently achieves the highest efficiency among all baselines com-

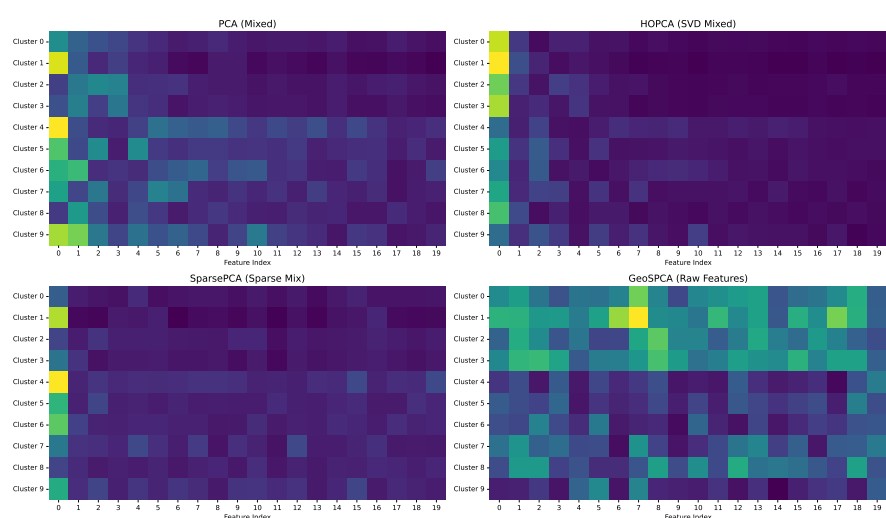

Figure 12: Comparison of semantic structure and interpretability reservation on MetaMathFewshot.

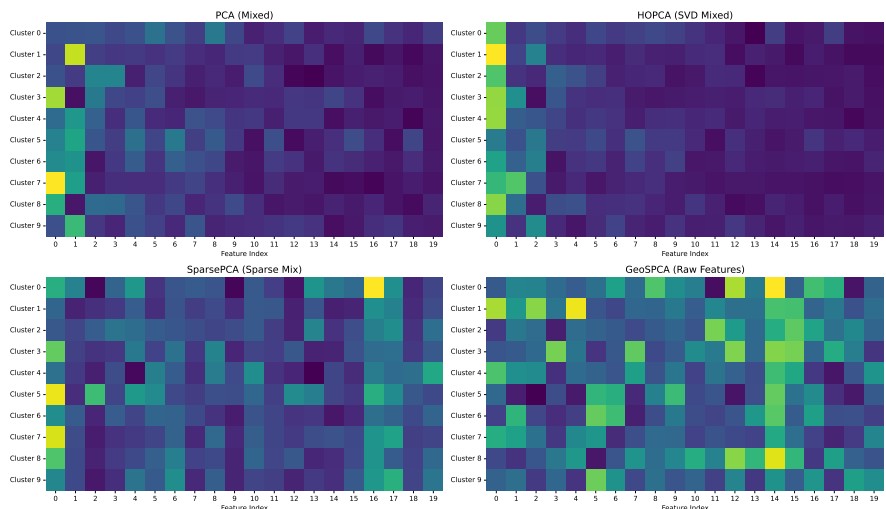

Figure 13: Comparison of semantic structure and interpretability reservation on CommonsenseQA.

Table 7: Comparison of latency (ms) of four methods across four datasets.

| Dataset | PCA | HOPCA | SparsePCA | GeoSPCA |
|---|---|---|---|---|
| GSM8K | 0.68 | 0.62 | 2.10 | **0.12** |
| AQuA-RAT | 4.22 | 3.81 | 3.54 | **0.17** |
| CSQA | 0.57 | 0.56 | 1.18 | **0.09** |
| MetaMathFewshot | 6.07 | 1.74 | 5.20 | **0.50** |

Table 8: Relative peak memory usage compared with GeoSPCA ($1.00\times$).

| Dataset | PCA | HOPCA | SparsePCA | GeoSPCA |
|---|---|---|---|---|
| GSM8K | $1.81\times$ | $1.00\times$ | $2.01\times$ | **$1.00\times$** |
| AQuA-RAT | $1.40\times$ | $1.00\times$ | $1.34\times$ | **$1.00\times$** |
| CSQA | $3.28\times$ | $3.26\times$ | $1.73\times$ | **$1.00\times$** |
| MetaMathFewshot | $1.10\times$ | $1.00\times$ | $2.11\times$ | **$1.00\times$** |

Table 9: Comparison of cluster consistency (ARI) across four datasets.

| Dataset | PCA | HOPCA | SparsePCA | GeoSPCA |
|---|---|---|---|---|
| GSM8K | **0.6930** | 0.5681 | 0.2042 | 0.2114 |
| AQuA-RAT | **0.7502** | 0.7240 | 0.3296 | 0.3500 |
| CSQA | **0.5123** | 0.4900 | 0.1065 | 0.1626 |
| MetaMathFewshot | **0.6223** | 0.5449 | 0.1913 | 0.2091 |

pared, which deliver acceleration **$5.7\times$–$24.8\times$** over PCA, **$3.5\times$–$22.4\times$** over HOPCA and **$10.4\times$–$20.8\times$** over SparsePCA as summarised in Table 7. In terms of memory consumption (Table 8), GeoSPCA also shows clear advantages and achieves a reduction in the peak memory of approximately **$1.10\times$–$3.28\times$** compared to PCA and **$1.34\times$–$2.11\times$** compared to SparsePCA, while maintaining a similar usage slightly above HOPCA. This is mainly because it bypasses the covariance computation and directly performs geometry-aware feature selection without iterative optimisation.

The experimental heatmap in Figure 4, Figure 11, Figure 12 and Figure 13 demonstrate that only GeoSPCA consistently produces interpretable cluster–feature corresponds that reflect preserved semantic structure in the embedding space across datasets of varying scales, ranging from GSM8K (5,000 samples) and CSQA (1,000 samples) to AQuA-RAT (10,000 samples) and MetaMathFewshot (40,000 samples), which across diverse semantic domains including arithmetic, symbolic, common-sense and mathematics. These results indicate that GeoSPCA reliably preserves semantic structure and interpretability even under extremely reduced dimensions ($K' = 20$ of 384) in high-dimensional sparse clustering scenarios. By contrast, GeoSPCA preserves essential semantic axes with feature-aware selection, whereas PCA, HOPCA and SparsePCA sacrifice partial or most semantic structure.

Additionally, reconstruction fidelity is typically less critical than semantic structural preservation and computational efficiency in this sparse, high-dimensional and large-scale scenario. As an expected limitation, GeoSPCA exhibits a tradeoff and consequently shows lower reconstruction fidelity than covariance-based methods such as PCA and HOPCA, as shown in Table 9. Even so, GeoSPCA consistently outperforms SparsePCA in all comparable settings because GeoSPCA retains raw semantically meaningful dimensions and benefits from stable feature-aware selection, while SparsePCA often converges to unstable local optima in structured high-dimensional spaces, resulting in weaker cluster separation and degraded semantic structure in dimension reduction case.

## J  INITIALIZATION AND TOLERANCE SENSITIVITY

We provide additional experiments to validate the effect of initialization strategies and tolerance parameter $\eta$ on sparse rank-$(1, 1, 1)$ tensors of size $50 \times 50 \times 50$. These results confirm that our solver is robust to initialization choices and stable under a wide range of tolerance parameters.

### J.1 SUPPORT SET INITIALIZATION

We compared three strategies for support set initialization:

1. **random**,
2. **top-norm** (largest $\ell_2$ columns),
3. **pca_loading** (first-PC loadings).

All achieved perfect reconstruction, but **top-norm** consistently minimized runtime and memory while preserving accuracy.

Table 10: Comparison of initialization strategies on sparse rank-$(1, 1, 1)$ tensors.

| Method | Time (s) | Memory (MB) | Error |
|---|---|---|---|
| random | $3.01 \times 10^{-2}$ | $2.50 \times 10^{-1}$ | $4.52 \times 10^{-12}$ |
| **top-norm** | $\mathbf{2.13 \times 10^{-2}}$ | $\mathbf{4.69 \times 10^{-3}}$ | $\mathbf{3.93 \times 10^{-12}}$ |
| pca_loading | $2.99 \times 10^{-2}$ | $5.05 \times 10^{-2}$ | $4.53 \times 10^{-12}$ |

### J.2 TOLERANCE PARAMETER $\eta$

In our implementation, $\eta$ is initialized once but dynamically updated: whenever a candidate support $s_0$ achieves $\eta(s_0) < \eta$, the tolerance is reset to $\eta(s_0)$ and enforced in subsequent iterations. A sensitivity study over $\eta \in [10^{-6}, 10^{-1}]$ (50 trials) shows stable error ($\approx 3.93 \times 10^{-12}$) and only minor runtime variation (0.020–0.023 s).

Table 11: Sensitivity of solver performance to different initial $\eta$.

| Initial $\eta$ | Total Error | Time (s) |
|---|---|---|
| $10^{-6}$ | $3.93 \times 10^{-12}$ | 0.022 |
| $3.59 \times 10^{-6}$ | $3.93 \times 10^{-12}$ | 0.023 |
| $1.29 \times 10^{-5}$ | $3.93 \times 10^{-12}$ | 0.021 |
| $4.64 \times 10^{-5}$ | $3.93 \times 10^{-12}$ | 0.021 |
| $1.67 \times 10^{-4}$ | $3.93 \times 10^{-12}$ | 0.021 |
| $5.99 \times 10^{-4}$ | $3.93 \times 10^{-12}$ | 0.021 |
| $2.15 \times 10^{-3}$ | $3.93 \times 10^{-12}$ | 0.021 |
| $7.74 \times 10^{-3}$ | $3.93 \times 10^{-12}$ | 0.021 |
| $2.78 \times 10^{-2}$ | $3.93 \times 10^{-12}$ | 0.020 |
| $10^{-1}$ | $3.93 \times 10^{-12}$ | 0.021 |

In practice, we recommend setting $\eta \in [10^{-4}, 10^{-2}]$ to ensure stable and efficient performance.

## K   DETAILS OF STARPLUS FMRI CLASSIFICATION EXPERIMENT

We evaluate our method on the StarPlus fMRI dataset (Just, 2002). For each human subject, the dataset contains 80 trials, where each trial corresponds to either reading a sentence or viewing an image. Each trial consists of 16 fMRI scans (time points sampled every 500 ms), and each scan is a 3D volume with 8 axial slices of size $64 \times 64$.

We arrange trials along the second mode to form a 5-way tensor:

$$(x, \text{ trials}, y, z, \text{ time}) = (64, 480, 64, 8, 16), \tag{21}$$

where each lateral slice along the trial mode represents one trial. All six subjects are included, producing 480 trials in total. We split the data into 90% training and 10% testing. Training trials are grouped by labels to construct two class tensors and compute mode-wise local bases (as described in Section 4.2). Each test trial is projected onto these bases, and *sparseGeoHOPCA* is used for

Table 12: Classification accuracy (%) on StarPlus fMRI data under different compression ratios.

| Compression Ratio | 1.0 (no compression) | 0.50 | 0.33 | 0.25 |
|---|---|---|---|---|
| Accuracy (%) | 85.92 | 75.56 | 71.85 | 70.37 |

compressed feature extraction. The StarPlus dataset contains 25–30 anatomically defined ROIs. We focus on the Left/Right Supramarginal Gyrus (LSGA/RSGA) regions for subjects 2 and 3 and evaluate classification accuracy under different compression ratios. The method maintains high recognition accuracy even with aggressive compression, as shown in Table 12.

These results indicate that the method preserves discriminative information in real high-dimensional fMRI data even under substantial compression.

## L   LARGE LANGUAGE MODELS USAGE STATEMENT

Large Language Models were only used to aid or polish writing.

