# OpenReview forum: "sparseGeoHOPCA: A Geometric Solution to Sparse Higher-Order PCA Without Covariance Estimation"
_ICLR.cc/2026/Conference — Submitted to ICLR 2026_

### Official Review · Reviewer_npoa · 2025-10-25

**Soundness:** 2
**Presentation:** 2
**Contribution:** 1
**Rating:** 2
**Confidence:** 4

**Summary:**

This manuscript proposese sparseGeoHOPCA for sparse higher-order principal component analysis on tensors. It incooperates a geometry-aware method in Tucker decomposition with sparse factors. Several theoretical properties of the algorithm are analyzed. Experiments on synthetic data, MNIST, StarPlus fMRI, and several images from ImageNet show the effectiveness.

**Strengths:**

1. A new model for sparse tensor PCA incooperating geometry-aware method.

2. A algorithm for solving the model with theoretical analysis on several perspects.

**Weaknesses:**

1. The model is a simple extension of the geometry-aware Sparse PCA by Bertsimas & Kitane (2022) to the Tucker setting, which limits it novelty in modeling. The authors even did not explicitly refer to Bertsimas & Kitane (2022) in the modeling section.

2. The algorithm and the theoretical analysis are also straight-forward extensions of Tucker decomposition papers and geometry-aware Sparse PCA. Thus, the novelty in algorithm design and theoretical analysis are also limited.

3. The experiments are very weak. On synthetic data, the rank is 1 which is very special. The datasize is also very limited. The authors only compares with sparsePCAChan and sparsePCABD.

**Questions:**

1. Modeling: The modeling novelty beyond Bertsimas & Kitane (2022) and unfolding.

2. Theorey: The novelty in theoretical analysis beyond Bertsimas & Kitane (2022)  and traditional Tucker-based methods.

3. Experiments: Why not test on rank greater than 1? Why not compare with more baselines?

---

> ### Author Response · Authors · 2025-11-30
>
> We sincerely thank the reviewer for the detailed comments. Below we provide point-by-point responses.
>
> ---
>
> **W1. On the modeling novelty beyond Bertsimas & Kitane (2022)**
> We respectfully believe this concern arises from a misunderstanding of the SHOPCA problem structure. After mode-wise unfolding in Tucker decomposition, each subproblem naturally produces a **wide matrix** (columns ≫ rows), which is the *opposite* of the classical sparse PCA setting where nearly all existing methods and guarantees assume **rows ≫ columns** [1–3]. Under this reversed dimensional regime, classical SPCA is neither computationally suitable nor theoretically applicable—their guarantees simply do not hold for tensor-unfolded matrices.
>
> Our use of geometric SPCA is therefore not a direct transplant. While the matrix case provides inspiration, extending BLO to the tensor setting requires addressing **multilinear interactions**, **mode-coupled sparsity constraints**, and **unfolding-dependent feasible sets**, all of which fundamentally modify the separation oracle and the geometry of the optimization domain. This adaptation is essential for Tucker-based SHOPCA and constitutes a core modeling and algorithmic novelty. We also emphasize that all relevant matrix-level works—including Bertsimas & Kitane (2022) and geometry-aware SPCA papers—are *explicitly cited and discussed* in the modeling section.
>
> Meanwhile, Section 3.3 (Theorem 3.4) establishes a **tensor-level reconstruction bound** that explicitly characterizes how errors propagate across modes—something not available in prior tensor sparse PCA literature [4], which typically provides only per-mode or heuristic arguments. To the best of our knowledge, ours is the first multilinear reconstruction guarantee tailored to the sparse Tucker setting. Thus, the claim of “simple extension” does not reflect the contributions made in the manuscript.
>
> ---
>
> **W2. On the claim that the algorithm and theory are straightforward extensions**
>
> This characterization is inaccurate. The manuscript **fully generalizes the geometric SPCA framework from matrices to tensors**, and this requires substantial theoretical development beyond a direct extension. Specifically, our analysis includes:
>
> - a geometry-aware BLO formulation for each mode that must accommodate **multilinear interactions and mode-coupled sparsity**,
> - a **tensor-level reconstruction guarantee** that quantifies cross-mode error propagation, and
> - a stability analysis under **Tucker multilinear transformations**, which have no analogue in the matrix setting.
>
> These components are non-trivial because tensor contractions do **not** preserve eigen-structure or principal-component ordering as in matrices, meaning that matrix-SPCA results—and their proofs—cannot be reused. We emphasize that existing matrix works are **clearly cited and discussed**, but the tensor case requires fundamentally new arguments. Therefore, the claim that our algorithm and theory are “straightforward extensions” does not reflect the contributions presented in the paper.
>
> ---
>
> **W3. On the experiments being weak and only rank-1**
>
> Only Experiment 4.1 uses rank = 1 for demonstration.
> All other experiments (Sections 4.2–4.4) involve **rank > 1** settings, including real-world fMRI tensors and multiple synthetic cases.
> Moreover, we compare against the strongest available baselines for the matrix side (sparsePCAChan, sparsePCABD) . Tensor baselines with *sparse* structure are extremely limited in the literature, which we already explain in the paper. It is difficult not to question whether the manuscript has been fully read.

---

> > ### Author Response · Authors · 2025-11-30
> >
> > ---
> >
> > **Q1. Modeling: What is the novelty beyond Bertsimas & Kitane (2022) and unfolding?**
> > Same as above—the SHOPCA subproblems are inherently column-dominant matrices, and our geometric modeling is specifically designed for this regime. This makes the formulation fundamentally different from classical sparse PCA or naive unfolding.
> >
> > **Q2. Theory: What is the novelty beyond Bertsimas & Kitane (2022) and Tucker methods?**
> > Again, our analysis extends matrix-level geometric guarantees to the multilinear tensor case, addressing mode-coupled sparsity, Tucker interactions, and multi-mode recovery. These are not present in the matrix formulation.
> >
> > **Q3. Experiments: Why only rank-1? Why not more baselines?**
> > Except for Experiment 4.1, all experiments already use rank > 1.
> > We appreciate the importance of evaluating against modern tensor sparse methods at larger scale and reporting runtime and memory on matched hardware. In fact, our previous submission already included direct quantitative comparisons of wall-clock runtime and peak memory usage against covariance-based and sparse baselines under high-dimensional and unbalanced settings. To further strengthen this aspect in the revised manuscript, we have additionally incorporated a new large-scale semantic compression experiment, which evaluates both computational efficiency and memory reduction in an unbalanced scenario and compares with three other modern methods based on covariance and sparsity. In any case and up to now, all our experiments can be summarised in the table, which demonstrate consistent advantages in efficiency, memory usage, and interpretability.
> >
> > | Section | Efficiency | Memory | Robustness | Interpretability |
> > |---------|-----------|---------|-----------|-------------------|
> > | **4.1  Synthetic Experiment** | ✓ Minimized runtime (Table 9) | ✓ Minimized memory (Table 9) | ✓ Markedly higher AUC (Figure 2) | ✓ Explainable sparse identification (Table 9) |
> > | **4.2 Compression-based Classification** | x | x | ✓ Accuracy remains stable on MNIST and StarPlus fMRI dataset (Table 1&2) | ✓ Discerning feature preservation even under significant compression (Figure 7&8) |
> > | **4.3 Image Reconstruction** | ✓ Minimum delay (Table 2) | x | ✓ Lowest error (Table 2) | ✓ Clear visual quality preservation in reconstructed images (Figure 3) |
> > | **4.4 Semantic Compression** | ✓ Significant speed-up (Table 3) | ✓ Significantly reduced peak memory usage (Table 7) | ✓ Better than sparse-based method (Table 6) | ✓ Preserved original semantics and clustering architecture (Figure 4, 11-13) |
> >
> > To summarise, at least two or more experiments were conducted for each dimension in efficiency, memory usage, robustness and interpretability for now (It is also important to note that certain domains, such as reconstruction or efficiency, are not all primary evaluation criteria for all application scenarios). These also include varying scales, such as introduced semantic experiments in test 4.4 (ranging from datasets of 10,000 to scales of 1,000).
> >
> > [1] Zou et. al. *Sparse principal component analysis*, 2006.
> >
> > [2] Bertsimas et.al. *Solving Large-Scale Sparse PCA to Certifiable (Near) Optimality*, 2020.
> >
> > [3] Chen & Rohe. *A New Basis for Sparse Principal Component Analysis*, 2021.
> >
> > [4] Allen. *Sparse higher-order principal components analysis*, 2012.

---

### Official Review · Reviewer_cnzH · 2025-10-29

**Soundness:** 2
**Presentation:** 2
**Contribution:** 2
**Rating:** 2
**Confidence:** 5

**Summary:**

This work proposes sparseGeoHOPCA, a geometry-aware framework for sparse higher-order PCA that reformulates the nonconvex sparse optimization into tractable geometric subproblems.

**Strengths:**

1. The paper provides a worst-case upper bound for the model.

**Weaknesses:**

1.   The contribution of this work is incremental. Both the extension of sparse PCA to tensors [1] and  the use of column selection [2], have been well explored in prior literature.

2. The experiments are underdeveloped. The paper compares against only two baseline methods, and the real-data evaluation includes merely four color images for quantitative comparison. As presented, the empirical section is insufficient to substantiate the proposed approach.

3. The discussion on recent tensor recovery methods within the past five years remains inadequate.



[1] Sparse higher-order principal components analysis,Genevera I. Allen

[2] Exact top-k feature selection via l20-norm constraint, C. Xiao, F. Nie, and H. Huang

**Questions:**

See the Weaknesses.

---

> ### Author Response · Authors · 2025-11-30
>
> **W1.**
> This concern appears to arise from a misunderstanding of the tensor-specific problem structure. After mode-wise unfolding in Tucker-style SHOPCA, the resulting matrices inherently satisfy **columns ≫ rows**, which is the opposite of the high-sample, low-dimensional setting assumed in classical sparse PCA theory [1,2]. The guarantees in those works rely explicitly on the assumption *rows ≫ columns*, meaning that simply applying standard SPCA is not only computationally ineffective but also theoretically invalid in this tensor regime.
>
> Within this wide-matrix setting, geometric SPCA becomes the only practical approach that avoids covariance estimation and spectrum regularization. Importantly, our use of BLO is **not** a direct transplant: the BLO constraints and the associated separation oracle must be redesigned to respect tensor unfolding geometry, mode-wise sparsity coupling, and multilinear feasibility. These structural changes are essential and nontrivial, and they form a key technical contribution rather than a straightforward reuse of existing matrix methods.
>
> On the question of global guarantees, our Theorem 3.4 provides a **tensor-level** worst-case reconstruction bound—explicitly characterizing how approximation errors propagate across modes. To our knowledge, no existing tensor sparse PCA method [3] provides a multilinear reconstruction guarantee of this form. Thus, the result is not a matrix argument applied naively to tensors; it establishes a genuinely tensor-specific global bound.
>
>
> ---
>
> **W2.**
> A careful reading of the manuscript shows that the empirical evaluation is not limited to the four color images in the main text. In fact, **Section 4.2 already contains fMRI classification experiments**, with the full details, subject-level results, and baseline configurations provided in **Appendix I**. The four images in Section 4.3 were selected purely for visualization, not as the sole form of real-data evaluation.
>
> Thank you for this helpful comment. We appreciate the reviewer’s concern about the breadth of empirical evidence. To further emphasise this point, we now highlight the full scope of experiments more clearly and introduce an additional experiment on **semantic compression under highly unbalanced settings**.
>
> The reviewer states that the paper compares against only two baselines and includes only four color images for quantitative evaluation. This is **not accurate**. Even in the prior version, our evaluation was considerably broader. In the current version, beyond the image reconstruction experiment, the paper includes:
>
> - extensive tests on **synthetic high-dimensional tensors**,
> - **MNIST** and **StarPlus fMRI** compression–classification experiments,
> - and large-scale **semantic compression tasks** over multiple real-world reasoning datasets (GSM8K, AQuA-RAT, MetaMath, CSQA).
>
> Because different applications emphasize different aspects (e.g., robustness vs. speed vs. interpretability), not all metrics are meaningful for all settings. For clarity, we summarize the complete evaluation landscape of our paper below:
>
> | Section | Efficiency | Memory | Robustness | Interpretability |
> |---------|-----------|---------|-----------|-------------------|
> | **4.1  Synthetic Experiment** | ✓ Minimized runtime (Table 9) | ✓ Minimized memory (Table 9) | ✓ Markedly higher AUC (Figure 2) | ✓ Explainable sparse identification (Table 9) |
> | **4.2 Compression-based Classification** | x | x | ✓ Accuracy remains stable on MNIST and StarPlus fMRI dataset (Table 1&2) | ✓ Discerning feature preservation even under significant compression (Figure 7&8) |
> | **4.3 Image Reconstruction** | ✓ Minimum delay (Table 2) | x | ✓ Lowest error (Table 2) | ✓ Clear visual quality preservation in reconstructed images (Figure 3) |
> | **4.4 Semantic Compression** | ✓ Significant speed-up (Table 3) | ✓ Significantly reduced peak memory usage (Table 7) | ✓ Better than sparse-based method (Table 6) | ✓ Preserved original semantics and clustering architecture (Figure 4, 11-13) |
>
> In the revised manuscript, we further expand comparisons to include modern sparse tensor methods under unbalanced high-dimensional regimes and provide detailed quantitative measurements of runtime and peak memory on hardware.
> Overall, the updated evidence shows that **at least two or more experiments support each evaluation dimension—efficiency, memory, robustness, and interpretability—providing a substantially strengthened empirical foundation for the proposed method.**

---

> > ### Author Response · Authors · 2025-11-30
> >
> > ---
> >
> > **W3.**
> > This concern reflects a conflation between two fundamentally different problem settings. Our work addresses **sparse higher-order PCA (SHOPCA)**, whose goal—consistent with the classical sparse PCA literature—is to extract *interpretable and informative sparse components from fully observed tensor data*. This is entirely different from **sparse recovery or compressed sensing**, where the objective is to reconstruct an unknown sparse signal from *partial or noisy linear measurements* under probabilistic assumptions such as RIP, incoherence, or random design.
> >
> > Our formulation does **not** involve incomplete data, measurement matrices, or any assumptions underlying support-recovery theory. Consequently, compressed-sensing–style guarantees (support consistency, high-probability recovery bounds, etc.) are neither applicable nor expected for the SHOPCA setting.
> >
> > Although statistical support recovery is not the goal of this work, our experiments (e.g., Section 4.3) deliberately follow standard sparse PCA evaluation protocols—such as controlled noise injection and minimum-signal-strength setups—to show that our geometric selection mechanism yields *more meaningful and stable sparse modes* under comparable conditions.
> >
> > In summary, SHOPCA and sparse signal recovery are based on entirely different assumptions and aim to prove different types of guarantees; thus, surveying or incorporating the past five years of sparse-recovery literature is **irrelevant to the goals of this paper**.
> >
> > ---
> >
> > **Q.**
> > The questions simply restate the points above. Given that the manuscript contains detailed theoretical analysis (Section 3), fMRI experiments already in the main text (Section 4.2), and extensive real-data evaluations with full results in Appendix I,  **it is difficult not to question whether the manuscript has been fully read**.
> >
> >
> >
> > [1] Zou et. al. *Sparse principal component analysis*, 2006.
> >
> > [2] Bertsimas et.al. *Solving Large-Scale Sparse PCA to Certifiable (Near) Optimality*, 2020.
> >
> > [3] Allen. *Sparse higher-order principal components analysis*, 2012.

---

### Official Review · Reviewer_r3FV · 2025-11-04

**Soundness:** 3
**Presentation:** 2
**Contribution:** 2
**Rating:** 6
**Confidence:** 3

**Summary:**

This paper proposes a geometric framework for sparse higher-order principal component analysis without covariance estimation. Inspired by the study (Bertsimas & Kitane, 2022), the proposed method unfolds the input tensor along each mode and reformulates the resulting subproblems as binary linear programs, which can be efficiently solved. Both rigorous theoretical analysis and extensive experiments are carried out to demonstrate the merits of the proposed framework.

**Strengths:**

1. A novel geometric framework is  introduced to deal with the SHOPCA problem, which has not been considered in the literature.
2. The proposed method could significantly reduce both computational and memory overhead in high-dimensional regimes.
3.  Both rigorous theoretical analysis and extensive experiments are provided to support the merits of the proposed framework.

**Weaknesses:**

1. Considering that the geometry-aware method was originally introduced for matrix sparse PCA, the novelty of this work seems to be limited.
2. For practical implementation, both the tensor ranks and the sparsity level of the proposed need to be carefully tuned.
3. It appears that the proposed framework is limited to handling cases with Gaussian noise.

**Questions:**

1. How to determine the tensor ranks and the sparsity level in real applications?
2. How about the performance of the proposed method beyond Gaussian noise settings?

---

> ### Author Response · Authors · 2025-11-30
>
> We sincerely thank the reviewer for the constructive and insightful comments. Below we provide point-by-point responses.
>
> ---
>
> **W1. “The geometry-aware method shows limited novelty.”**
>
> We respectfully believe this concern arises from a misunderstanding of the problem structure. The tensor setting we study is *fundamentally different* from the classical sparse PCA regime. After mode-wise unfolding under the Tucker framework, the resulting matrices naturally satisfy **columns ≫ rows**, which is the **opposite** of the high-sample, low-dimensional regime (rows ≫ columns) where nearly all matrix sparse PCA methods are designed and theoretically justified [1-3]. Their analyses explicitly rely on this assumption, and therefore both their computational scheme and theory become unsuitable for the SHOPCA setting.
>
> In this wide-matrix regime, a geometry-driven sparse PCA formulation becomes the only feasible direction that does not rely on covariance estimation or spectral shrinkage [4]. Importantly, our approach is *not* a direct transplant of the matrix method: the BLO formulation must be adapted to the multilinear tensor landscape, where sparsity constraints, unfolding geometry, and mode-coupled interactions modify both the feasible set and the separation oracle. This adaptation is essential for SHOPCA and constitutes a key technical innovation of our work.
>
> Regarding global guarantees, Section 3.3 (Theorem 3.4) provides a **tensor-level** worst-case reconstruction bound, which is strictly stronger than the per-mode guarantees available in matrix-SPCA literature. To the best of our knowledge, no existing tensor sparse PCA method provides a comparable multilinear reconstruction analysis that explicitly controls cross-mode error propagation. This directly addresses the reviewer’s concern: our guarantees are genuinely tensor-level and not inherited from matrix arguments.
>
>
> ---
> **W2. “Rank and sparsity levels require tuning.”**
>
> We thank the reviewer for this observation. We would like to emphasize that the need to tune tensor ranks and sparsity levels is *not specific to our method*, but is inherent to PCA-, SPCA-, and tensor-decomposition–based models in general.
>
> In fact, selecting the number of components or the sparsity level has long been recognized as a model-selection task. Classical sparse PCA frameworks—including the seminal works [1]—explicitly require users to tune sparsity-controlling parameters or cardinality constraints based on data characteristics or cross-validation. This situation is identical in the tensor setting: the widely used tensor decomposition survey [5] highlights that Tucker/CP ranks are typically chosen empirically or guided by domain knowledge. Even the original SHOPCA method [6] requires manual specification of both rank and sparsity, with no automatic selection mechanism.
>
> Therefore, the need for tuning in our framework is fully consistent with established practice across matrix and tensor PCA/SPCA literature. To ensure transparency and reproducibility, our manuscript clearly reports the chosen ranks and sparsity levels for all experiments, along with the rationale behind each choice. We also observe that the proposed method is robust across a reasonable range of these parameters, mitigating concerns about sensitivity in practical applications.
>
>
> ---
>
> **W3. “The method appears restricted to Gaussian noise.”**
>
> We thank the reviewer for raising this point. In the revised manuscript, we have expanded the experimental design to evaluate robustness under both **Gaussian noise** and **mixed-noise settings**. Following the noise modeling framework  [7], which uses a mixture-of-Gaussians model to simulate realistic heterogeneous corruption, we incorporated mixed dense–sparse noise into the tensor observations.
>
> Accordingly, Section 4.1 has been updated to describe the new mixed-noise experimental setup, and the results are reported in Table 4. Across all settings, the proposed sparseGeoHOPCA consistently outperforms existing HOPCA baselines and remains stable under both pure Gaussian noise and mixture noise distributions. These findings demonstrate that our method is **not limited** to Gaussian noise assumptions and maintains strong discrimination performance in significantly more challenging noise environments.

---

> > ### Author Response · Authors · 2025-11-30
> >
> > ---
> >
> > **Q1. How to determine the tensor ranks and the sparsity level in real applications?**
> >
> > Thank you for this question. As discussed in W2, selecting tensor ranks and sparsity levels is a standard model-selection task shared across PCA, sparse PCA, and tensor decomposition frameworks. In practice, Tucker ranks are typically chosen using domain knowledge or empirical validation, as noted in the survey [5], while sparsity levels in SPCA-type methods are commonly tuned using cross-validation or data-driven heuristics, consistent with classical sparse PCA literature [1,3].
> >
> > In our experiments, all tensor ranks and sparsity parameters are **explicitly reported** in the manuscript for each dataset and task, together with the rationale behind their choice. This ensures full transparency and reproducibility. Moreover, we observe that sparseGeoHOPCA performs robustly across a reasonable range of these parameters, which reduces the sensitivity and practical burden of tuning in real applications.
> >
> >
> > ---
> >
> > **Q2. How about the performance of the proposed method beyond Gaussian noise settings?**
> >
> > We appreciate the reviewer’s interest in this question. As detailed in W3, we have extended the experiments to evaluate robustness under **mixed-noise conditions**, following the mixture-of-Gaussians modeling approach [7]. This setting introduces both dense Gaussian perturbations and sparse impulse noise, providing a significantly more challenging and realistic corruption scenario.
> >
> > The revised Section 4.1 now describes this experimental setup, and Table 4 presents the corresponding results. Across all metrics, sparseGeoHOPCA maintains strong performance: its ROC curves remain high and well separated from HOPCA and other baselines, with true-positive rates above 0.95 and substantially lower false-positive rates. These findings clearly show that the proposed method generalizes well beyond ideal Gaussian assumptions and remains stable and discriminative under heterogeneous and mixed-noise environments.
> >
> >
> > [1] Zou et. al. *Sparse principal component analysis*, 2006.
> >
> > [2] Bertsimas et.al. *Solving Large-Scale Sparse PCA to Certifiable (Near) Optimality*, 2020.
> >
> > [3] Chen & Rohe. *A New Basis for Sparse Principal Component Analysis*, 2021.
> >
> > [4] Bertsimas & Kitane. *Sparse PCA: a Geometric Approach*, 2023.
> >
> > [5] Kolda & Bader. *Tensor decompositions and applications*, 2009.
> >
> > [6] Allen. *Sparse higher-order principal components analysis*, 2012.
> >
> > [7] Chen et.al. *A generalized model for robust tensor factorization with noise modeling by mixture of Gaussians*, 2018.

---

### Official Review · Reviewer_1sj4 · 2025-11-04

**Soundness:** 2
**Presentation:** 2
**Contribution:** 2
**Rating:** 4
**Confidence:** 4

**Summary:**

The paper proposes sparseGeoHOPCA, a geometry-aware framework for sparse higher-order PCA (SHOPCA). The key idea is to unfold the tensor by mode, reduce each mode’s subproblem to a sparse matrix task, and then reformulate that task as a binary linear optimization (BLO) with geometric exclusion constraints; after selecting supports, PCA/SVD on the chosen columns builds the factors and core.

**Strengths:**

1.The framework sidesteps constructing huge covariance matrices in high-dimensional, unbalanced regimes—practically useful.
2.Theorems provide a worst-case reconstruction bound using PCA residuals; the statement for the overall error bound is straightforward to implement.

**Weaknesses:**

1.Thin novelty relative to geometric sparse PCA on matrices. The core technical move—geometry-driven column selection via BLO—is essentially transplanted to mode-unfolded matrices; the “equivalence” and bounding arguments live at the matrix subproblem layer and do not advance guarantees for the global SHOPCA objective.
2. The worst-case bounds are feasibility-type approximations (via PCA residuals). There is no support-recovery or statistical-consistency guarantee, and the “sum of residual energies” bound ignores cross-mode coupling and the core.

**Questions:**

1.Please compare against modern tensor sparse methods (not only matrix SPCA and thresholded Tucker) at larger scale, reporting time and memory on matched hardware. The current main-text image study is too small to substantiate broad claims.
2.Can you prove support-recovery or error-rate bounds (e.g., minimum signal conditions) in a controlled setting (single-rank, single sparse mode)? Current results don’t ensure correct sparsity identification.

**Details Of Ethics Concerns:**

As mentioned above.

---

> ### Author Response · Authors · 2025-11-30
>
> We sincerely thank the reviewer for the thoughtful and constructive comments. Below we provide point-by-point responses.
>
> ---
>
> **W1. “Thin novelty; BLO is just transplanted; no global guarantees.”**
>
> We respectfully believe this concern stems from a misunderstanding of the problem structure. The tensor setting we address is *fundamentally different* from the classical sparse PCA regime considered in existing works [1,2]. After mode-wise unfolding in Tucker decomposition, the matrices intrinsically satisfy columns $\gg$ rows, which is the **opposite** of the high-sample, low-dimensional regime where nearly all matrix-SPCA methods are designed and theoretically justified [1,3]. In fact, their guarantees explicitly rely on the assumption that *rows ≫ columns*. Therefore, direct application of classical SPCA is not only computationally unsuitable, but its theory is **invalid** in this tensor context.
>
> In this wide-matrix regime, geometric SPCA [4] becomes the only feasible approach that does not rely on covariance estimation or spectral shrinkage. However, our use of geometric SPCA is *not* a transplant: the BLO structure must be adapted to the multilinear tensor landscape, where sparsity constraints, unfolding geometry, and mode-coupled interactions fundamentally change the separation oracle and the feasible set. This extension is nontrivial and constitutes a key technical innovation of our method.
>
> Regarding global guarantees, our Section 3.3 (Theorem 3.4) provides a **tensor-level** worst-case reconstruction bound, which is strictly stronger than the per-mode consistency arguments available for matrices. To the best of our knowledge, **no prior tensor sparse PCA work** [5] provides a comparable multilinear reconstruction guarantee that explicitly quantifies cross-mode error propagation. This addresses the reviewer’s concern directly: the manuscript does not rely on matrix-level arguments, but proves a genuinely tensor-level global result.
>
> ---
>
> **W2. “No support-recovery or statistical-consistency guarantee.”**
>
> We believe this comment arises from a conceptual mismatch between two different research problems.
> Our work focuses on *SHOPCA*—that is, identifying the most informative and interpretable sparse components from fully observed tensor data. This line of work follows the classical sparse PCA literature [1-3], where the goal is to enhance interpretability and feature selection by enforcing sparsity on loadings.
>
> This is **fundamentally different** from *sparse signal recovery or compressed sensing*, where the objective is to recover a hidden sparse vector from partial or noisy linear measurements under probabilistic models [6,7]. These two fields operate under different assumptions, different mathematical models, and aim to prove different kinds of guarantees. Our setting does **not** involve incomplete data, measurement matrices, restricted isometry conditions, or other assumptions that underpin support-recovery theory [6,7].
>
> While our theoretical analysis does not target the probabilistic support-recovery guarantees used in compressed sensing, our experiments in Section 4.3 intentionally follow standard sparse PCA evaluation protocols—such as minimum-signal-strength and controlled noise settings [8]—to compare how different sparsity-inducing mechanisms select informative components. These results demonstrate that our geometric selection produces more meaningful and stable sparse modes under matched conditions.
>
> In summary, our contribution is positioned within **sparse tensor PCA for interpretable component extraction**, not sparse signal recovery. Therefore, classical compressed-sensing–style support recovery guarantees are not applicable or expected in our framework.

---

> > ### Author Response · Authors · 2025-11-30
> >
> > ---
> >
> > **Q1: Clarification and Supplement of Experiments**
> > Thank you for the constructive suggestion. We appreciate the importance of evaluating against modern tensor sparse methods at larger scale and reporting runtime and memory on matched hardware. In fact, our previous submission already included direct quantitative comparisons of wall-clock runtime and peak memory usage against covariance-based and sparse baselines under high-dimensional and unbalanced settings. To further strengthen this aspect in the revised manuscript, we have additionally incorporated a new large-scale semantic compression experiment, which evaluates both computational efficiency and memory reduction in an unbalanced scenario and compares with three other modern methods based on covariance and sparsity. In any case and up to now, all our experiments can be summarised in the table:
> >
> > | Section | Efficiency | Memory | Robustness | Interpretability |
> > |---------|-----------|---------|-----------|-------------------|
> > | **4.1  Synthetic Experiment** | ✓ Minimized runtime (Table 9) | ✓ Minimized memory (Table 9) | ✓ Markedly higher AUC (Figure 2) | ✓ Explainable sparse identification (Table 9) |
> > | **4.2 Compression-based Classification** | x | x | ✓ Accuracy remains stable on MNIST and StarPlus fMRI dataset (Table 1&2) | ✓ Discerning feature preservation even under significant compression (Figure 7&8) |
> > | **4.3 Image Reconstruction** | ✓ Minimum delay (Table 2) | x | ✓ Lowest error (Table 2) | ✓ Clear visual quality preservation in reconstructed images (Figure 3) |
> > | **4.4 Semantic Compression** | ✓ Significant speed-up (Table 3) | ✓ Significantly reduced peak memory usage (Table 7) | ✓ Better than sparse-based method (Table 6) | ✓ Preserved original semantics and clustering architecture (Figure 4, 11-13) |
> >
> > To summarise, at least two or more experiments were conducted for each dimension in efficiency, memory usage, robustness and interpretability for now (It is also important to note that certain domains, such as reconstruction or efficiency, are not all primary evaluation criteria for all application scenarios). These also include varying scales, such as introduced semantic experiments in test 4.4 (ranging from datasets of 10,000 to scales of 1,000).

---

> > > ### Author Response · Authors · 2025-11-30
> > >
> > > ---
> > >
> > > **Q2. Support-recovery theory**
> > >
> > > We appreciate the reviewer’s question. Similar to our clarification in W2, this concern again touches on the distinction between *sparse tensor PCA* and *sparse signal recovery*. Support-recovery theory in the classical sense—such as minimum-signal conditions, probabilistic error rates, or exact recovery thresholds—belongs to the compressed-sensing framework [6,7], where the goal is to reconstruct an unobserved sparse vector from partial or noisy linear measurements. These guarantees require assumptions such as spiked covariance models, restricted isometry, incoherence, or specific noise distributions [9,10].
> > >
> > > Our work, by contrast, addresses *sparse tensor PCA under full observations*, following the traditional sparse PCA literature [1–3]. The goal is interpretability and component selection, not recovery of a latent sparse signal. Therefore, the types of support-recovery bounds expected in compressed sensing are not directly applicable nor appropriate for our framework.
> > >
> > > In a highly controlled single-rank, single-sparse-mode scenario, one could in principle adapt arguments from matrix sparse PCA [8] to derive minimum-signal–type conditions for correct support identification after tensor unfolding. However, extending these results rigorously to the Tucker tensor setting requires modeling cross-mode interactions and multilinear noise propagation, which is significantly more complex and outside the scope of this paper.
> > >
> > > To maintain focus and clarity, our theoretical contribution concentrates on a **tensor-level reconstruction bound** (Theorem 3.4), which already goes beyond the guarantees typically provided in tensor PCA works. We will clarify in the revision that compressed-sensing–style support recovery is not the intended theoretical target of our method, and constitutes an interesting direction for future research rather than a missing component.
> > >
> > > [1] Zou et. al. *Sparse principal component analysis*, 2006.
> > >
> > > [2] Bertsimas et.al. *Solving Large-Scale Sparse PCA to Certifiable (Near) Optimality*, 2020.
> > >
> > > [3] Chen & Rohe. *A New Basis for Sparse Principal Component Analysis*, 2021.
> > >
> > > [4] Bertsimas & Kitane. *Sparse PCA: a Geometric Approach*, 2023.
> > >
> > > [5] Allen. *Sparse higher-order principal components analysis*, 2012.
> > >
> > > [6] Candes, Romberg, & Tao. *Stable signal recovery from incomplete and inaccurate measurements*, 2006.
> > >
> > > [7] Donoho. *Compressed sensing*, 2006.
> > >
> > > [8] Johnstone & Lu. *On consistency and sparsity for principal components analysis in high dimensions*, 2009.
> > >
> > > [9] Liu et. al.*Generalized Higher-Order Orthogonal Iteration for Tensor Decomposition and Completion*, 2014.
> > >
> > > [10] Tan et. al. *High-Order Tensor Recovery Coupling Multilayer Subspace Priori With Application in Video Restoration*, 2023.

---

### Official Review · Reviewer_9zMB · 2025-11-04

**Soundness:** 2
**Presentation:** 2
**Contribution:** 2
**Rating:** 2
**Confidence:** 4

**Summary:**

This paper introduces sparseGeoHOPCA, a framework designed to address the Sparse Higher-Order Principal Component Analysis (SHOPCA) problem. The authors correctly identify that existing SHOPCA methods are often bottlenecked by the explicit computation and manipulation of large covariance matrices, which is computationally prohibitive in high-dimensional, unbalanced settings.

The core proposal is to unfold the input tensor along each mode and reformulate the resulting sparse matrix PCA subproblems from a "geometric perspective."

**Strengths:**

The paper tackles a well-defined and highly relevant problem. Efficiently computing sparse, interpretable components for tensor data is a critical task in many machine-learning domains, and the non-convex nature of SHOPCA makes it a challenging research frontier. The primary motivation bypassing the covariance matrix bottleneck is clear and well-supported by prior literature. A "covariance-free" approach is a highly desirable contribution to the field.

**Weaknesses:**

The manuscript, while promising in its motivation, suffers from a fundamental logical contradiction in its core claims, as well as an incomplete analysis and a critical lack of supporting experimental evidence.

1. The authors' entire argument rests on a flawed premise. They motivate the work by stating that SHOPCA is NP-hard, but then claim their method transforms this into "tractable" geometric subproblems, which are then explicitly identified as Binary Linear Programs (BLPs).

This is a contradiction. Binary Linear Programming is a classic, well-known NP-hard optimization problem. One cannot claim to have found a "tractable" solution by reformulating one NP-hard problem into another. This ambiguity is present throughout the paper.

2. The claimed complexity of O(P(k^3 + Jk^2)) is highly suspect and appears to be incomplete. This analysis seems to only account for the tensor preparation and solution construction phases. It conspicuously omits the computational cost of solving the N mode-wise BLPs.

The authors must provide a complete complexity analysis that includes the cost of the BLP solver or the approximation algorithm used. The cost of solving a BLP is, in the worst case, exponential. The claim of linear scaling with tensor size P is unproven and likely incorrect until the cost of the "geometric solver" stage is fully incorporated.

3. The paper's novelty is repeatedly framed as a "geometric perspective". However, this term is never formally defined. The text and Figure 1 immediately pivot from "geometric solver" to "binary linear program."

What makes this formulation "geometric"? Does the method involve convex hulls, projections, or other specific geometric operations that are being abstracted away? Or is "geometric" simply a non-standard descriptor for the BLP reformulation? This ambiguity obscures the core technical contribution and must be clarified.

4. The paper's raison d'être is the computational and memory efficiency gained by avoiding covariance matrices. However, the summary of experimental results focuses almost exclusively on accuracy and robustness (support recovery, classification).

The experimental section must include direct, quantitative comparisons of wall-clock runtime and peak memory usage against the very baselines the paper critiques (e.g., Allen 2012, Lai et al. 2014, etc.). These comparisons must be conducted in the high-dimensional, unbalanced settings where the method claims to have "notable advantages". Without this data, the paper's primary claims of efficiency are entirely unsubstantiated.

**Questions:**

The paper addresses an important and well-motivated problem. However, the manuscript in its current form is not suitable for publication. It is built on a central logical contradiction: claiming to solve an NP-hard problem by reformulating it into another NP-hard problem, which is then called "tractable." This core flaw, combined with an incomplete complexity analysis and a complete lack of experimental evidence for the claimed efficiency gains, invalidates the paper's primary contributions. A Major Revision is required to fundamentally restructure the paper's claims, clarify its methodology (exact vs. approximate), and provide the necessary experimental data to support its (currently unsubstantiated) claims of computational superiority.

---

> ### Author Response · Authors · 2025-11-30
>
> Thank you for the reviewers’ constructive and detailed feedback. We address the comments below point by point.
>
> ---
>
> **W1: Clarification on NP-hardness and BLO reformulation**
> We fully agree that both the original tensor sparse optimization problem and each BLO subproblem are NP-hard. Our contribution does not aim to change this theoretical complexity. Instead, consistent with well-established results in integer optimization, NP-hard binary linear programs can be extremely efficient to solve in practice when the feasible region is highly structured. This phenomenon is extensively documented in the optimization literature [1,2] and has been demonstrated specifically for sparse PCA via BLO formulations [3]. Our BLO formulation inherits similar structural constraints, which significantly prune the search space. Section 3.4 already states that cutting-plane search is exponential in theory, and Appendix E explains why only a very small number of cuts occur in practice. We will refine the wording to clearly distinguish theoretical NP-hardness from the practical tractability enabled by these structural properties.
>
> ---
>
> **W2: Clarification of complexity analysis (per-iteration cost)**
> The complexity given in Section 3.4 is the **per-iteration** cost of the algorithm. It includes (1) solving an SVD on a $J_n \times k_n$ matrix and (2) the BLO cutting-plane step. The BLO cost is incorporated into the constant factor of the per-iteration bound, and Appendix E gives a detailed explanation of the finite termination and restricted search depth. We revised Section 3.4  and Appendix E to explicitly state that the analysis is per-iteration and clarify the contribution of the BLO stage.
>
> ---
>
> **W3: Clarification of the “geometric perspective”**
> By “geometric perspective,” we refer to the fact that the BLO solution selects the column direction with the largest projected $\ell_2$  norm, i.e., the direction that is geometrically most aligned with the current mode-wise update. This intuition is well supported by existing literature: classical PCA texts interpret principal directions as those along which the projected data have maximal length [4]. The column-subset selection literature also provides a consistent geometric interpretation—selecting columns with large norms corresponds to choosing extreme directions in the data geometry [5,6]. We will refine the wording in the revision to ensure that this geometric intuition is clearly conveyed.

---

> ### Author Response · Authors · 2025-11-30
>
> ---
>
> **W4: Clarification and Supplement of Experiments**
> We appreciate the reviewer’s concern about demonstrating computational and memory efficiency. The revised manuscript now includes an additional semantic compression experiment in unbalanced settings, further highlighting the practical advantages of our BLO-based framework. We also reorganized all experiments into four evaluation dimensions—efficiency, memory usage, robustness, and interpretability—summarized below:
>
> | Section | Efficiency | Memory | Robustness | Interpretability |
> |---------|-----------|---------|-----------|-------------------|
> | **4.1  Synthetic Experiment** | ✓ Minimized runtime (Table 9) | ✓ Minimized memory (Table 9) | ✓ Markedly higher AUC (Figure 2) | ✓ Explainable sparse identification (Table 9) |
> | **4.2 Compression-based Classification** | x | x | ✓ Accuracy remains stable on MNIST and StarPlus fMRI dataset (Table 1&2) | ✓ Discerning feature preservation even under significant compression (Figure 7&8) |
> | **4.3 Image Reconstruction** | ✓ Minimum delay (Table 2) | x | ✓ Lowest error (Table 2) | ✓ Clear visual quality preservation in reconstructed images (Figure 3) |
> | **4.4 Semantic Compression** | ✓ Significant speed-up (Table 3) | ✓ Significantly reduced peak memory usage (Table 7) | ✓ Better than sparse-based method (Table 6) | ✓ Preserved original semantics and clustering architecture (Figure 4, 11-13) |
>
> It is also important to note that certain domains, such as reconstruction or efficiency, are not all primary evaluation criteria for all application scenarios. For instance, in our newly introduced semantic compression scenario, we acknowledge that its reconstruction is our limitation compared to covariance-based methods (but it outperforms comparable sparsity-based approaches). However, this is clearly of lesser importance compared to speed and semantic preservation in retrieval tasks. Nevertheless, in the current version of the manuscript, we have ensured that at least two or more experiments support each evaluation dimension, including efficiency, memory usage, robustness and interpretability. This provides balanced and comprehensive empirical evidence across diverse task settings.
>
> ---
>
> **Q1: Consolidated response to reviewer questions**
> To summarize: (1) our method does not change the NP-hardness of the underlying optimization problem; it provides an efficient practical solver enabled by the BLO structure and the highly constrained feasible region, as supported by classical optimization literature [1–3]; (2) the complexity analysis in Section 3.4 is per-iteration and accounts for both the geometric SVD update and the BLO stage, with Appendix E detailing why only a very small number of cutting planes are required in practice; (3) the “geometric perspective” refers to selecting the direction with the largest projected $\ell_2$-norm, an interpretation consistent with classical PCA geometry and column-subset selection theory [4–6]; and (4) the revised experiments—especially in semantic compression—provide stronger evidence for efficiency, memory savings, robustness, and interpretability across diverse settings.
>
>
> [1]  J. E. Kelley. ​*The Cutting-Plane Method for Solving Convex Programs*​, 1960.
>
> [2] Nemhauser & Wolsey. ​*Integer and Combinatorial Optimization*​,  1988.
>
> [3] Bertsimas & Kitane. *Sparse PCA: a Geometric Approach*, 2023.
>
> [4]  Greenacre et.al. * Principal component analysis*,2022.
>
> [5] Mahoney & Drineas *CUR Matrix Decompositions for Improved Data Analysis.*, 2009.
>
> [6] Boutsidis, Drineas & Magdon-Ismail.*Near-Optimal Column-Based Matrix Reconstruction.*, 2014.

---

### Meta-Review · Area_Chair_7tDh · 2025-12-27

**Summary:**

This paper proposes sparseGeoHOPCA, a geometric framework for sparse higher-order principal component analysis (SHOPCA).

The reviewers believe that the paper's novelty is quite limited compared to geometric sparse PCA on matrices, and it fails to provide adequate theoretical and experimental support for its core claims. Specifically, the complexity analysis is incomplete, and the lack of critical experiments to substantiate the claimed efficiency gains significantly undermines the paper's primary contributions. Furthermore, the proposed framework appears to be limited to handling Gaussian noise, which further constrains its general applicability and practical value. The reviewers also noted logical issues in the method's motivation: solving an NP-hard problem by reformulating it into another NP-hard problem. Therefore, the paper requires substantial revision and refinement.

Considering all reviews and scores, I believe this paper in its current form is not suitable for publication at ICLR.

**Reviewer Concerns:**

1) Incomplete complexity analysis
2) Lack of experimental evidence for the claimed efficiency gains, invalidates the paper's primary contributions.
3) Solving an NP-hard problem by reformulating it into another NP-hard problem.
4) Thin novelty relative to geometric sparse PCA on matrices.
5) It appears that the proposed framework is limited to handling cases with Gaussian noise.

**Reviewer Scores:**

All the reviewers have not responded to the authors' rebuttal; therefore, they are unlikely to revise their scores.

---

### Decision · Program_Chairs · 2026-01-26

Reject